# Variable-Length Audio Fingerprinting

## Abstract

Audio fingerprinting converts audio to much lower-dimensional representations, allowing distorted recordings to still be recognized as their originals through similar fingerprints. Existing deep learning approaches rigidly fingerprint fixed-length audio segments, thereby neglecting temporal dynamics during segmentation. To address limitations due to this rigidity, we propose Variable-Length Audio Finger-Printing (VLAFP), a novel method that supports variable-length fingerprinting. To the best of our knowledge, VLAFP is the first deep audio fingerprinting model capable of processing audio of variable length, for both training and testing. Our experiments show that VLAFP outperforms existing state-of-the-arts in live audio identification and audio retrieval across three real-world datasets.

## 1 Introduction

With the rapid growth of digital broadcasting, advertisers have an increasing need to verify that their commercials are aired as contracted (Zhang et al., 2025; He et al., 2025). Consequently, audio fingerprinting has gained increasing research attention (Su et al., 2024; Cortès et al., 2022). Audio fingerprinting maps an audio signal to a compact, low-dimensional representation (Burges et al., 2005), enabling applications such as summarization, deduplication, and identification (Cotton & Ellis, 2010; Hon et al., 2015; Chen et al., 2024). These applications often rely on a retrieval framework, where a reference audio database is first created, and then query audios are fingerprinted and matched against it to retrieve the most similar entries. In broadcast monitoring, for example, fingerprints of commercials are stored in a reference database, and any aired audio is fingerprinted and compared against the database to verify whether a contracted commercial has been broadcast.

Two properties, *robustness* and *reliability*, are central to the effectiveness of audio fingerprinting (Haitsma & Kalker, 2002). For an audio signal $a$, robustness requires that the fingerprint of a distorted version $a'$ remains similar to that of $a$. This ensures robustness to audio degradation. Conversely, reliability requires fingerprints of unrelated audios to be separated, which guarantees correct retrieval than mismatches. To achieve these properties, earlier methods extracted salient acoustic features to fingerprint audios (Wang, 2003). With the success of deep acoustic models, recent research has shifted to deep learning for more meaningful representations (Araz et al., 2025).

Nevertheless, existing deep audio fingerprinting methods remain *hindered by* their reliance on fixed-length segmentation. We highlight three critical limitations that motivate a departure from fixed-length segmentation: (1) **Loss of Natural Boundaries.** Fixed-length segmentation often cuts across semantic or acoustic boundaries, splitting words, phrases, or musical notes, and therefore fails to capture coherent audio units. (2) **Redundant or Noisy Context.** Segments of fixed-length may contain pure silence or irrelevant sounds, resulting in noisy segments and wasted computation on less informative portions. (3) **Distortion Incompatibility.** Fixed-length segmentation misaligns segments under certain audio distortions, particularly time-stretching. Fig. 1 shows these limitations.

Existing audio fingerprinting methods cannot simply switch to variable-length segmentation, as they are only designed for fixed-length segments. Hence, we propose a novel method named Variable-Length Audio FingerPrinting (VLAFP), which fingerprints audio of arbitrary and continuous variable lengths and thereby addresses these limitations. VLAFP uses metric learning to embed an audio signal close to its distortions and far from unrelated audios. Built on a transformer backbone with stacked self-attention and cross-attention layers, VLAFP captures inter-frame relations within segments and learns segment-level representations across frames. The final aggregated embedding serves as the audio fingerprint, and is trained with contrastive learning loss.

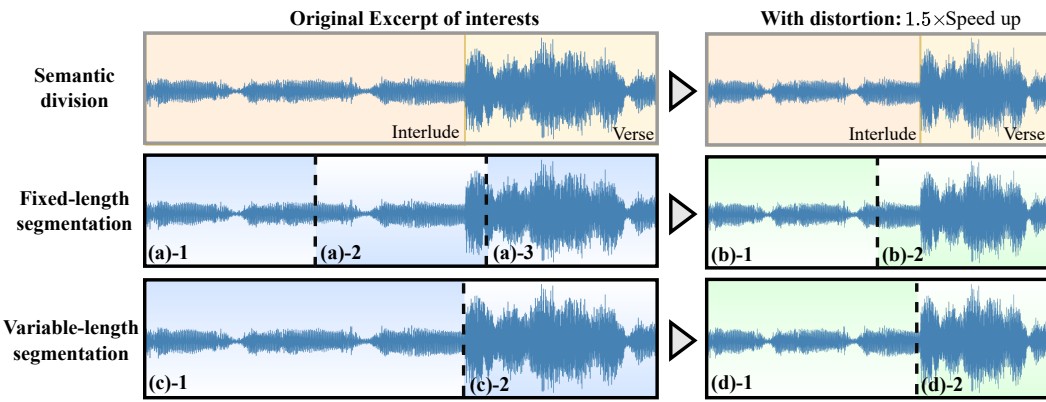

Figure 1: Limitations of fixed-length audio fingerprinting. The top row shows a 3-second excerpt (containing both interlude and verse) in its original form and with a $1.5\times$ speed-up. The middle and bottom rows compare fixed-length segmentation and variable-length segmentation. Fixed-length segmentation suffers from three issues: **Loss of Natural Boundaries.** Segments cut across semantic units ((a)-2), complicating interpretation. **Redundant or Noisy Context.** Segments oversimplify ((a)-1) or combine too much information ((a)-2). **Distortion Incompatibility.** Time-stretch prevents exact matching; no subfigure in (b) aligns perfectly with (a)'s. Variable-length segmentation overcomes these issues by producing segments aligned with semantic boundaries. Our proposed VLAFP leverages variable-length segmentation and addresses these limitations.

We evaluate VLAFP on live audio identification and offline audio retrieval. Leveraging its variable-length capability, VLAFP is trained and tested on segments with distortions, including time stretching, background mixing, and impulse response convolution. Results show that VLAFP learns robust and reliable fingerprints for both tasks. Our contributions are summarized below,

- To the best of our knowledge, we are the first to propose a deep variable-length audio fingerprinting method, *Variable-Length Audio FingerPrinting (VLAFP)* along with a variable-length segmentation method, to address multiple limitations of fixed-length fingerprinting.
- Experiments show that VLAFP consistently outperforms existing methods on live audio identification and offline audio retrieval across three real-world datasets, which opens up multiple directions for future research, paving the way for future work on segmentation strategies, data augmentations, and self-supervised loss functions.

## 2 RELATED WORK

***Statistical Audio Fingerprinting.*** Earlier methods extracted salient features as audio fingerprints. For example, local maxima in the time-frequency representation can serve as fingerprints (Wang, 2003; Worldveil, 2013). These local maxima are often referred to as peaks, and collectively as landmarks or constellations. Landmark-based methods are robust against background noise in audio identification, since noise tends to have lower intensity in the time-frequency representation. However, it is difficult to determine the required number of points, which can grow rapidly when the audio length increases. Other methods fingerprint audio using principal component analysis on the audio spectrogram (Agarwaal et al., 2023), or select features based on filters and quantizers (Jang et al., 2009). These approaches also fail to identify audio when time stretching is present.

***Deep Audio Fingerprinting.*** More recently, deep learning methods have been developed for audio fingerprinting (Chang et al., 2021; Su et al., 2024). These methods aim to learn embeddings for audio segments such that the embeddings of an audio signal and its distortions lie close together in the embedding space. Several models leverage CNN encoders to learn fingerprints from fixed-length spectrograms (e.g., one second) (Bhattacharjee et al., 2025; Singh et al., 2022). To improve efficiency, (Su et al., 2024) further utilizes a transformer to aggregate 1-second segments to a coarser unit, such as 10-second segments. More recent work studies the audio fingerprinting effectiveness.

For example, (Araz et al., 2025) investigates the effectiveness of audio fingerprinting under different contrastive learning loss functions. These methods are all limited by fixed-length segmentation.

***Variable-length Acoustic Models.*** Deep variable-length acoustic modeling has been widely studied across various tasks, including speaker verification, emotion recognition, and speech translation, among others (Kim et al., 2022; Hsu et al., 2021; Baevski et al., 2020; Zhang et al., 2023; Pagnoni et al., 2024). (Kim et al., 2022) proposes a neural network with multi-scale layers to identify speakers. (Baevski et al., 2020) utilizes transformers to encode audio sequences for character or phoneme prediction. (Hsu et al., 2021) proposes a transformer-based model that predicts cluster assignments of masked speech frames. Different from the target tasks of these methods, our VLAFP distinctively tackles audio fingerprinting.

## 3 VARIABLE-LENGTH AUDIO FINGERPRINTING

### 3.1 PROBLEM FORMULATION

Given an audio signal $a = a[n]$ consisting of $n$ samples, where $n$ denotes a variable length, we aim to train a fingerprinting model $f : a \rightarrow \mathbf{z} \in \mathbb{R}^d$. Our proposed VLAFP maps the audio segment to a $d$-dimensional fingerprint, denoted by $\mathbf{z}$. The goal is that for any distortion $a'$ of $a$, its fingerprint $\mathbf{z}' = f(a')$ is close to $\mathbf{z}$, i.e., $\mathbf{z} \approx \mathbf{z}'$. Correspondingly, our objective function $f$ is: $f = \mathrm{argmax}_{f_\phi} \mathbb{E}_a \left[ f_\phi(a)^\mathsf{T} f_\phi(a') \right]$, with $\phi$ denoting the model parameter. Hence, VLAFP enables retrieval of the original recording even when the query is distorted.

### 3.2 VARIABLE-LENGTH DUAL-ATTENTION TRANSFORMER

VLAFP builds upon four types of layers commonly used in transformer-based models: normalization layers (RMSNorm), self-attention layers (SelfAttn), cross-attention layers (CrossAttn), and feedforward networks (FFN). For clarity, we provide their definitions with a notation table in Appendix A. To overcome the limitations of fixed-length segmentation, we design five novel steps that integrate these layers into the overall VLAFP architecture, as illustrated in Fig. 2: (a) initial projection, (b) inter-frame self-attention, (c) frame-to-segment cross-attention, (d) segment embedding initialization, and (e) fingerprint summarization.

***Initial Projection.*** We first transform signal $a$ into its time-frequency representation to obtain its spectrogram, denoted by $\mathbf{A} \in \mathbb{R}^{T \times F}$, where $T$ is the number of audio frames, and $F$ is a selected number of frequency bins. Note that $T$ is variable since it is proportional to the number of samples $n$. Each element in $\mathbf{A}$ represents the intensity at the corresponding time frame and frequency. VLAFP treats the number of frequency bins $F$ as the initial feature dimension $d_0 = F$ and projects the spectrogram to $d_1$ dimensions through a linear layer, $h^0 = \mathbf{A} W_0 + b_0 \in \mathbb{R}^{T \times d_1}$. The projection layer is followed by a stack of $L$ transformer blocks, where each block contains in sequence a self-attention layer and a cross-attention layer.

***Inter-frame Self-Attention.*** The self-attention layers aim to learn the inter-relationships among time frames. Let $\tilde{h}^l$ denote the frame-level embeddings at block $l$, computed as

$$h^l = h^{l-1} + \mathrm{SelfAttn}_l \left( \mathrm{RMSNorm} \left( h^{l-1} \right) \right) \in \mathbb{R}^{T \times d_2} \tag{1}$$

$$\tilde{h}^l = h^l + \mathrm{FFN}_l \left( \mathrm{RMSNorm} \left( h^l \right) \right) \in \mathbb{R}^{T \times d_2} \tag{2}$$

Intuitively, self-attention layers enable each audio frame to integrate information from other frames, while projecting the representations to dimension $d_2$. Inter-frame learning effectively enhances the robustness of the learned representations in our variable-length setting by integrating more consistent information across frames within each segment than traditional fixed-length methods.

***Frame-to-segment Cross-Attention.*** Cross-attention layers aggregate frame-level representations into segment-level representations by taking segment embeddings $s$ as the query vector and frame embeddings $\tilde{h}$ as the key and value vectors. We use a superscript to indicate them for different blocks. In this way, cross-attention layers learn how each frame attends to segment embeddings. Let

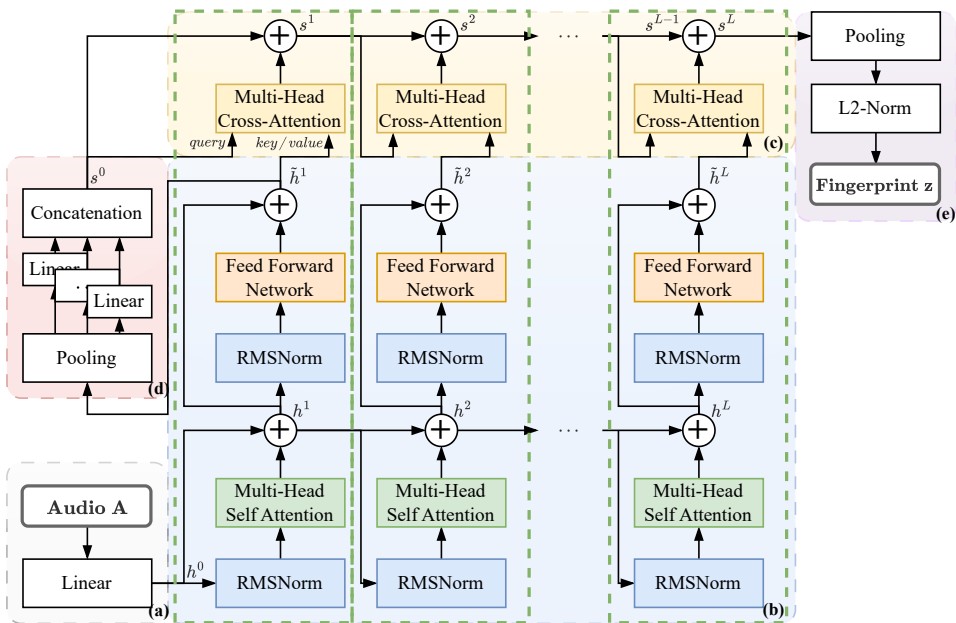

Figure 2: The architecture of our proposed VLAFP model. (a) **Initial Projection:** Audio **A** in its spectrogram representation is projected through a linear layer. (b) **Inter-frame Self-Attention:** Multi-head self-attention layers learn inter-frame relationships. (c) **Frame-to-segment Cross-Attention:** Multi-head cross-attention layers model the frame-to-segment relationships. (d) **Segment Embedding Initialization:** Replicas of segment embeddings are initialized through frame-to-segment pooling. (e) **Fingerprint Summarization:** Replicas of segment embeddings are aggregated and L2-normalized to generate an audio fingerprint **z**.

$x_q^l$ denote the query input and $x_{kv}^l$ denote the key and value input to the cross-attention layer in the $l$(th) block. Let $H$ denote the number of segment embeddings in a block. Let $s^l$ denote the segment embeddings in the $l$(th) block. We select $x_q^l = s^{l-1} \in \mathbb{R}^{H \times d}$ as the $H$ segment embeddings from the $l-1$(th) block. We select $x_{kv}^l = \tilde{h}^l \in \mathbb{R}^{T \times d}$ as the frame embeddings at $l$(th) block. Hence, the segment embedding at the $l$(th) block is computed as

$$s^l = s^{l-1} + \text{CrossAttn}\left(x_q^l, x_{kv}^l\right) \tag{3}$$

$$= s^{l-1} + \text{CrossAttn}\left(s^{l-1}, \tilde{h}^l\right) \in \mathbb{R}^{H \times d} \tag{4}$$

Note that the $T$ frames reduce to one segment for each of the $H$ embeddings.

***Segment Embedding initialization.*** For the first cross-attention layer at block 1, VLAFP initializes segment embeddings $s^0$ through pooling and projection of the frame embeddings $\tilde{h}^1$:

$$s_h^0 = \text{Pooling}\left(\tilde{h}^1\right) W_s \in \mathbb{R}^d, \qquad s^0 = \left[s_1^0; s_2^0; \ldots; s_H^0\right] \in \mathbb{R}^{H \times d} \tag{5}$$

where a pooling function (e.g., mean) aggregates all frame embeddings in $\tilde{h}^1$ to a vector of dimension $\mathbb{R}^{d_2}$, which is then projected with $W_s \in \mathbb{R}^{d_2 \times d}$. We derive one embedding per head, denoted as $s_h^0$, and concatenate all $H$ embeddings to form the initial query embeddings $s^0$ for the cross-attention layer. Via pooling-based aggregation and concatenation, VLAFP maps variable-length inputs (of $T$ frames) into a unified dimensionality in the transformer.

***Fingerprint Summarization.*** VLAFP applies mean pooling to the final segment embeddings $s^L$ along the embedding dimension ($H$) and normalizes the result to generate a fingerprint **z**:

$$\mathbf{z} = \underset{L2}{\text{Norm}}\left(\text{Pooling}\left(s^L\right)\right) \in \mathbb{R}^d \tag{6}$$

---

**Algorithm 1:** Pseudo Code for Variable-Length Audio Segmentation. (Details in Algorithm 2)

---

**1 Input:** audio signal $a$, thresholds $T_{\min}, T_{\max}, \theta$
**2 while** *audio $a$ remains* **do**
**3**     1. Initialize with $T_{\min}$ frames and compute entropy stats
**4**     2. Extend while $T < T_{\max}$ and entropy remains stable
**5**     3. Stop when z-score $> \theta$; emit segment and continue
**6 end**

---

Overall, VLAFP fingerprints audio segments of arbitrary length (either $n$ of $a$ or $T$ of $\mathbf{A}$) by modeling inter-frame relationships through self-attention, capturing frame-to-segment relationships through cross-attention, and subsequently summarizing the resulting embeddings into the final fingerprint. Since these layers integrate dynamic information across frames, VLAFP learns more robust and reliable representations for audio fingerprinting.

### 3.3 OBJECTIVE

We adopt supervised contrastive learning, which allows multiple positive and negative samples per anchor in a batch $\mathcal{B}$, thereby enhancing robustness and reliability of fingerprints (Khosla et al., 2020).

$$
\mathcal{L}(\mathcal{B}) = -\sum_{a \in \mathcal{B}} \frac{1}{|P(a)|} \sum_{a^+ \in P(a)} \log \frac{\exp(\mathbf{z} \cdot \mathbf{z}^+ / \tau)}{\sum_{a^* \in \mathcal{B} \setminus a} \exp(\mathbf{z} \cdot \mathbf{z}^* / \tau)}, \qquad \begin{cases} \mathbf{z}^+ &= f_{\text{VLAFP}}(a^+) \\ \mathbf{z}^* &= f_{\text{VLAFP}}(a^*) \end{cases} \tag{7}
$$

The objective iterate over a batch $\mathcal{B}$ where at each iteration a sample $a \in \mathcal{B}$ is designated as the anchor. It then averages over the positive samples of $a$, denoted by $a^+ \in P(a)$. Let $\mathbf{z} = f(a)$ denote the fingerprint of $a$. The objective amplifies the similarity between the fingerprints of the anchor, $\mathbf{z}$, and of its positive samples, $\mathbf{z}^+$, relative to the total similarity between $\mathbf{z}$ and all other samples $\mathbf{z}^*$. $\tau$ denotes a temperature parameter.

## 4 EXPERIMENTAL SETUP

***Datasets & Baselines.*** We experiment on three widely used datasets covering music, speech, and general audio: *Free Music Archive (FMA)*, *LibriSpeech*, and *AudioSet* (Defferrard et al., 2018; Panayotov et al., 2015; Gemmeke et al., 2017). We compare VLAFP with two deep fingerprinting methods, *NAFP* and *AMG* (Chang et al., 2021; Su et al., 2024). Moreover, we compare with three general audio representation learning methods, including *wav2vec2*, *HuBERT*, and *AST* (Baevski et al., 2020; Hsu et al., 2021; Gong et al., 2021).

***Variable-Length Segmentation.*** We propose a novel variable-length method based on spectral entropy, as described in Algorithm 1. Spectral entropy is an audio feature that measures the uncertainty in the frequency intensity distribution. For a given audio frame, its spectral entropy is computed by (1) applying a short-time Fourier transform to obtain energy concentrations at different frequencies, (2) normalizing these frequency energies as a distribution, and (3) applying Shannon's entropy formula to this distribution (Misra et al., 2004). Intuitively, low spectral entropy indicates a concentrated frequency distribution, as exemplified by pure tones. High spectral entropy indicates a uniform frequency distribution, as observed in white noise. Our method maintains a window and determines whether the next audio frame has spectral entropy close to the average in the window, using a *z-score threshold* $\theta$. Based on the evaluation, the window either expands or is finalized as a segment, and a new window is initiated with the current frame. Segment lengths are constrained between a minimum and a maximum, $[T_{\min}, T_{\max}]$. A key strength of our variable-length segmentation is that it subsumes fixed-length segmentation as special cases. When $\theta = 0$, no new frame is added to current window, and our method reduces to fixed-length segmentation with the minimum length $T_{\min}$. Conversely, when $\theta = +\infty$, all frames are admitted, yielding fixed-length segments of the maximum length $T_{\max}$. For intermediate values, our method produces variable-length segments, allowing users to tune $\theta$ to favor short or long segments.

Table 1: Setup of *Commercial-Broadcast Retrieval (CBR)* and *Dummy-Target Retrieval (DTR)*.

| Task | Fingerprint database source | Query (distorted) | Segmentation | | Distortion | | | Objective | Dataset | Metric |
|------|------------------------------|-------------------|------|-------------|----|----|----|-----------|---------|--------|
| | | | VL | FL (1 sec) | TS | BG | IR | | | |
| *CBR* | a commercial | distorted broadcast | VLAFP | baselines | ✓ | ✓ | ✓ | identify commercial | all | precision, recall, F1 |
| *DTR* | dummy + original target | distorted target | | all methods | | ✓ | ✓ | retrieve original | *FMA* | top-1 hit rate |

Table 2: CBR results on *FMA*, *LibriSpeech*, and *AudioSet*. Values are reported as percentages (%) for *precision*, *recall*, and *F1-score*, with the best scores highlighted in **bold** and second best in *italics*.

| Method | FMA | | | LibriSpeech | | | AudioSet | | |
|--------|-----------|--------|------|-----------|--------|------|-----------|--------|------|
| | Precision | Recall | F1 | Precision | Recall | F1 | Precision | Recall | F1 |
| wav2vec2 | 5.76 | 32.78 | 9.79 | 3.88 | 26.09 | 6.75 | 5.32 | **40.63** | 9.40 |
| HuBERT | 4.87 | **98.94** | 9.29 | 4.53 | 15.48 | 7.01 | 6.96 | 25.25 | 10.91 |
| AST | 9.62 | 30.88 | 14.68 | 3.80 | 25.03 | 6.59 | 12.80 | 26.24 | 17.21 |
| AMG | 25.81 | 31.37 | 28.32 | 22.51 | 28.82 | 25.28 | 17.48 | 27.72 | 21.44 |
| NAFP | *75.84* | 64.23 | *69.55* | *44.74* | *33.37* | *38.23* | **55.95** | 34.32 | *42.54* |
| **VLAFP (Ours)** | **81.00** | *70.15* | **75.19** | **50.19** | **44.06** | **46.93** | *49.58* | *39.17* | **43.75** |

***Audio Augmentation & Time-frequency Representation.*** To create positive samples for training, we augment segments with a chain of time-stretching (**TS**), background noise mixing (**BG**), impulse response convolution (**IR**). We then apply a mel-spectrogram transformation to both the original segments and their augmentations.

***Tasks - Commercial-Broadcast Retrieval & Dummy-Target Retrieval.*** We validate the effectiveness of VLAFP on two tasks: Commercial-Broadcast Retrieval (**CBR**) and Dummy-Target Retrieval (**DTR**). Both tasks rely on a vector database (Douze et al., 2024).

CBR aims to identify a commercial of interest within a broadcast. Hence, after segmentation and fingerprinting, we construct a fingerprint database from the commercial. We then simulate a broadcast containing the commercial and therefore know its location. Next, we segment and fingerprint the broadcast (assume resulting $K$ segments) and query each broadcast segment to retrieve the most similar commercial segment in terms of inner product score. The broadcast is segmented and fingerprinted, resulting in $K$ segments (where $K$ depends on the segmentation procedure), and each broadcast segment is queried against the commercial database to retrieve the most similar segment based on inner product score. This produces $K$ pairs of ⟨broadcast segment, retrieved commercial segment⟩ and their associated scores. A threshold on the inner product score determines whether a broadcast segment is identified as the commercial. Since the ground truth is known, we can compute True Positives (TP), False Positives (FP), and False Negatives (FN) for any threshold, and thereby derive Precision $= \frac{\text{TP}}{\text{TP+FP}}$, Recall $= \frac{\text{TP}}{\text{TP+FN}}$, and F1 $= \frac{2 \cdot \text{Precision} \cdot \text{Recall}}{\text{Precision+Recall}}$. Since the optimal threshold varies across methods, we report results using the threshold that maximizes F1.

Conversely, DTR is inspired by copyrighted song detection. DTR aims to retrieve the correct target audio from a large database using fingerprints of distorted target audios (i.e., songs with distortion). The database is built on both unrelated audios (dummy) and audios of interest (target). Each query corresponds to a distorted version of a target audio. For a query of duration $k$ seconds, DTR fingerprints $2k-1$ segments (using a 1-second window with a 0.5-second hop) and retrieves $2k-1$ segments from the database. Since these $2k-1$ retrieved segments may come from different audios, we identify the audio that contributes the majority of retrieved segments, and designate that as the retrieved audio. For each query, the Top-1 Hit is 1 if the correct audio is retrieved and 0 otherwise. The *Top-1 Hit Rate* is the average of these Top-1 Hit values across all queries. Table 1 summarizes the setups for both tasks, including segmentation, distortion, etc.

Appendix B provides additional details for datasets and baselines (Appendix B.1), variable-length segmentation (Appendix B.2), audio augmentation and time-frequency representation (Appendix B.3), and both tasks of Commercial-Broadcast Retrieval (CBR) and Dummy-Target Retrieval (DTR) (Appendix B.4), along with other training configurations (Appendix B.5).

Table 3: DTR results on FMA. Top-1 Hit Rate is reported for different methods and query durations.

| Method | Number of Seconds in Query on *FMA* (Top-1 Hit Rate) | | | | | |
|---|---|---|---|---|---|---|
| | 1 | 2 | 3 | 5 | 6 | 10 |
| wav2vec2 | 0.10 | 0.05 | 0 | 0 | 0 | 0 |
| HuBERT | 0.10 | 0.15 | 0.05 | 0.10 | 0.05 | 0 |
| AST | 1.65 | 3.55 | 5.25 | 8.90 | 10.15 | 16.60 |
| AMG | 11.05 | 21.20 | 30.15 | 41.40 | 45.00 | 55.05 |
| NAFP | *53.85* | *79.95* | *89.70* | **96.10** | *97.25* | *99.15* |
| **VLAFP** | **59.55** | **84.40** | **91.30** | *96.00* | **97.30** | *99.20* |

Table 4: A comparison of model size, training and inference efficiency.

| Method | Params. (M) | Train (sec / epoch) | Inference (ms / seg) |
|---|---|---|---|
| wav2vec2 | 94.4 | - | **6.4** |
| HuBERT | 315.5 | - | 12.2 |
| AST | 86.6 | - | 34.1 |
| AMG | **4.4** | **285.6** | 6.7 |
| NAFP | 16.9 | 1186.6 | 22.8 |
| **VLAFP** | 12.2 | 773.9 | 78.2 |

Table 5: A comparison of storage across methods for CBR and DTR on the FMA dataset. The best (lowest) values are highlighted in **bold** and second best in *italics*.

| Method | dim $d$ | CBR Commercial | | CBR Broadcast | | DTR Dummy | | DTR Target | |
|---|---|---|---|---|---|---|---|---|---|
| | | #seg | size | #seg | size | #seg | size | #seg | size |
| wav2vec2 | 1568 | 15k | 87M | 289k | 1.7GB | 581k | 3.6GB | 30k | 177MB |
| HuBERT | 1568 | 15k | 87M | 289k | 1.7GB | 581k | 3.6GB | 30k | 177MB |
| AST | 527 | 15k | 29M | 289k | 581MB | 581k | 1.3GB | 30k | 60MB |
| AMG | 128 | 15k | **7M** | 289k | **141MB** | 581k | **299MB** | 30k | **15MB** |
| NAFP | 128 | 15k | **7M** | 289k | **141MB** | 581k | **299MB** | 30k | **15MB** |
| **VLAFP (Ours)** | 256 | **12k** | *12M* | **220k** | *214MB* | 581k | *597MB* | 30k | *29MB* |

# 5 RESULTS

## 5.1 COMMERCIAL-BROADCAST RETRIEVAL

Table 2 reports the precision, recall, and F1 scores for each method and dataset in the Commercial-Broadcast Retrieval (CBR) task. (1) VLAFP achieves the best or second-best performance metrics across all methods. Specifically, VLAFP has the best precision, recall, and F1 score on LibriSpeech, the best precision on FMA, the best F1 score on both FMA and AudioSet, and remains competitive elsewhere. In contrast, general audio representational learning approaches (wav2vec2, HuBERT, AST) yield F1 scores below $20\%$, which indicates their limited use for CBR. (2) VLAFP significantly outperforms the baselines on LibriSpeech. Intuitively, VLAFP likely benefits most from variable-length segmentation on LibriSpeech, which avoids forming segments that cross speech-silence boundaries and thus produces more coherent fingerprints. (3) HuBERT exhibits an extreme imbalance, with near-perfect recall ($98.94\%$) but very low precision, giving an F1 of only $7.01\%$. This pattern suggests that HuBERT indiscriminately classifies most segments as commercials (hence recalling all of them in the broadcast), which nullifies its practical use.

## 5.2 DUMMY-TARGET RETRIEVAL

For Dummy-Target Retrieval (DTR), we follow the setup in NAFP (Chang et al., 2021) and use *FMA* to construct a fingerprint database from $10,000$ unrelated audios (dummy) and $500$ audios of interest (target), for a total of $10,500$ audios. We then apply distortions to the target audios to create query audios. Each query audio is evaluated with multiple durations: $\{1, 2, 3, 5, 6, 10\}$ seconds. Table 3 reports the Top-1 Hit Rate for different methods and query durations. Our proposed VLAFP consistently outperforms the baselines. Notably, for $1$-second queries, VLAFP achieves a $+5\%$ improvement over the best baselines, and it maintains higher Top-1 Hit Rates with longer durations.

## 5.3 MODEL SIZE, RUNTIME, AND STORAGE EFFICIENCY

Table 4 compares model size, training time, and inference time. VLAFP has fewer parameters than most baselines: with 12.2 M parameters, it is 28% smaller than NAFP (16.9 M). It also trains faster than NAFP, which demonstrates improved efficiency. Although the AMG method achieves the smallest model size and shortest training time, its performance across metrics is much worse (by 12–60%). The longer inference time of VLAFP is due to overhead from loading and locating variable-length segments in the *PyTorch Dataloader*, since we adopt masking to indicate positions of each segment within a data row. For fixed-length segmentation, each segment is 1 second which

Table 6: Ablation study with CBR.

| Method | VLAFP on *FMA* | | |
|---|---|---|---|
| | Precision | Recall | F1 |
| VLAFP | **81.00** | **70.15** | **75.19** |
| −w/o Self | 65.30 | 63.65 | 64.47 |
| −w/o Cross | 74.87 | 67.02 | 70.73 |

Table 7: Ablation study with DTR.

| Method | Number of Seconds in Query on FMA (Top-1 Hit Rate) | | | | | |
|---|---|---|---|---|---|---|
| | 1 | 2 | 3 | 5 | 6 | 10 |
| VLAFP | **59.55** | **84.40** | **91.30** | **96.00** | **97.30** | **99.20** |
| −w/o Self | 51.90 | 76.10 | 86.10 | 93.20 | 94.90 | 97.95 |
| −w/o Cross | 51.35 | 75.55 | 85.55 | 93.30 | 94.55 | 97.75 |

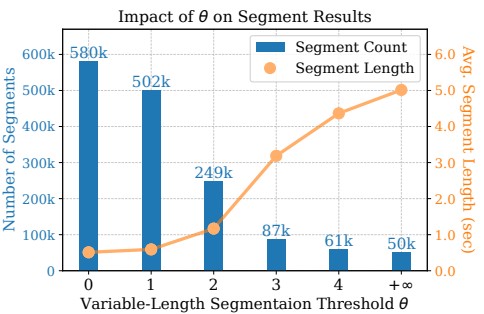

Figure 3: Segment count and average length.

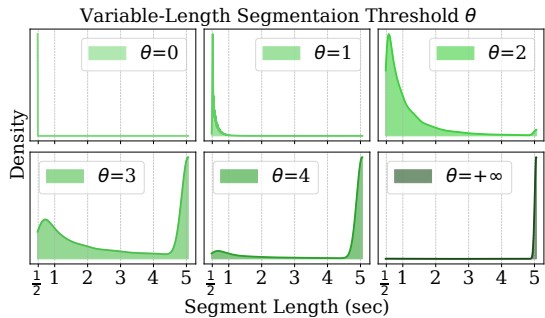

Figure 4: Segment length distribution.

makes loading straightforward. For variable-length segmentation, segments vary in duration from 0.5 to 5 seconds. We apply data packing, where multiple shorter segments can be packed into a single row. Masks are used to indicate the boundaries of each segment within the row. Notably, this will not be a bottleneck since the inference time (78.2 ms) is much smaller than the minimum segment length (0.5 seconds).

Table 5 compares the storage efficiency in the query stage for both CBR and DTR tasks. As observed, The required storage size grows linearly to both (1) fingerprint dimension $d$ and (2) number of segments #seg. For the CBR task, our proposed spectral entropy-based segmentation derives fewer segments, and hence increase storage efficiency. For the DTR task, since all methods leverage fixed-length segmentation of (1 second), the number of segments is the same across methods. The required storage is proportional to the fingerprint dimension $d$. This implies researchers can easily improve storage efficiency by selecting a small $d$ (e.g., $d = 128$ instead of our $d = 256$).

## 5.4 ABLATION STUDY

We conduct an ablation study to investigate the effectiveness of the self-attention layers and cross-attention layers in VLAFP. Specifically, we create VLAFP variants by removing the self-attention layer (w/o Self) and the cross-attention layer (w/o Cross) to evaluate their individual contributions to CBR and DTR tasks. Both variants show degraded performance on the two tasks, highlighting the importance of these attention mechanisms. Notably, CBR performance significantly deteriorates without self-attention, as shown in Table 6. The F1 score drops from 75.19% to 64.47% (a 10%+ relative decrease), indicating that CBR requires extensive frame-level information mining and temporal modeling within audio sequences. When cross-attention are removed, CBR performance also has a ~5% decrease on F1 score. For DTR, both layers contribute similarly to performance, as shown in Table 7. The Top-1 Hit Rate drops ~8% on duration of 1 second, indicating their effectiveness.

## 5.5 IMPACTS OF HYPERPARAMETERS

***Impact of Threshold $\theta$ in Variable-Length Segmentation.*** We experiment with various *z-score segmentation threshold* $\theta \in \{0, 1, 2, 3, 4, +\infty\}$. As shown in Fig. 3, as $\theta$ increases from 0 to $+\infty$, the number of segment decreases while the average segment length increases. Specifically, our segmentation reduces to a fixed-length of $T_{\min} = 0.5$ under $\theta = 0$ and $T_{\max} = 5$ under $\theta = +\infty$. Fig. 4 shows how $\theta$ affects the distribution of resulting segment lengths. For both CBR and DTR tasks, we observe better results with smaller $\theta$, as shown in Table 8 and Fig. 5. This indicates that VLAFP learns more effectively from shorter segments. Note that VLAFP has near-100% hit rates

Table 8: CBR results of VLAFP on FMA when trained with different segmentation threshold $\theta \in \{0, 1, 2, 3, 4, +\infty\}$. Higher values indicate better performance. Smaller $\theta$ (0 or 1) yields better results, which corresponds to shorter segments.

| Segmentation | VLAFP on *FMA* | | |
|---|---|---|---|
| Threshold $\theta$ | Precision | Recall | F1 |
| 0 | 79.28 | **71.63** | **75.26** |
| 1 | **81.00** | 70.15 | 75.19 |
| 2 | 74.86 | 65.24 | 69.72 |
| 3 | 74.16 | 61.31 | 67.12 |
| 4 | 71.99 | 64.95 | 68.29 |
| $+\infty$ | 71.94 | 68.68 | 70.27 |

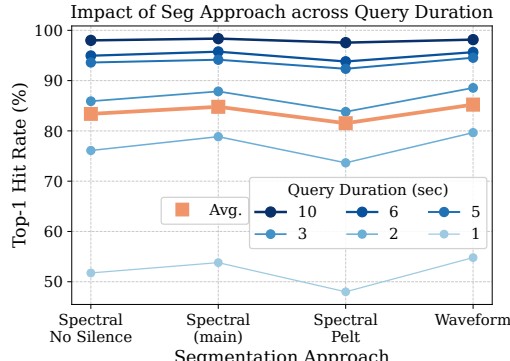

Figure 5: DTR results of VLAFP on FMA with different $\theta$ configurations.

Table 9: CBR results of VLAFP on FMA when trained with different segmentation methods, including our current segmentation method (*main*) and three other variable-length segmentation methods. Higher values indicate better performance. VLAFP has the best results on the *main* segmentation.

| Segmentation | VLAFP on *FMA* | | |
|---|---|---|---|
| Method | Precision | Recall | F1 |
| (*main*) | **81.00** | **70.15** | **75.19** |
| No Silence | 76.29 | 67.86 | 71.83 |
| Pelt | 69.84 | 60.31 | 64.73 |
| Waveform | 78.1 | 68.60 | 73.07 |

Figure 6: DTR results of VLAFP on FMA with different segmentation methods.

at a query duration of 10 seconds. This means that even though VLAFP is trained on 5-second segments ($\theta = \infty$), it can still perform robustly on shorter, overlapping segments at test time. Moreover, shorter segments from small $\theta$ allows VLAFP to preserve more frame-level details. The tradeoff is longer runtime time and increased storage.

***Impact of Segmentation Methods.*** Our proposed VLAFP enables new research opportunities for incorporating any variable-length segmentation methods. Building on our current variable-length segmentation method (named as *main*), we further propose and evaluate three alternative methods: (1) *No Silence* removes all silence (defined as 60 dB below the peak) before applying the baseline segmentation. (2) *Pelt* uses a change point detection algorithm to segment audio based on changes in spectral entropy (Killick et al., 2012). (3) *Waveform* is similar to our main approach except it applies the *main* method to the waveform entropy rather than spectral entropy. For both tasks, our main segmentation approach yields the best performance, as shown in Table 9, while for the DTR task, Waveform also has competitive performance, as shown in Fig. 6.

***Impact of Augmentation Types.*** We investigate the impact of different audio augmentations used in training. We train a model instance for each of the $2^{|\{TS,BG,IR\}|}$ combinations of three augmentations: Time Stretching (TS), Background Mixing (BG), Impulse Response Convolution (IR). For the CBR task, IR is the most effective augmentation in comparison with the other two augmentation types. The best performance is achieved when all augmentations are used, as shown in Table 10. For the DTR task, both BG and IR contribute significantly to performance, as shown in Fig. 7.

***Impact of Number of Positive Samples per Anchor Sample.*** We experiment with different numbers of positive samples per anchor, denoted as #pos $\in \{1, 2, 3, 4, 5\}$. For the CBR task, the best F1 score

Table 10: CBR results of VLAFP on FMA with different augmentations: time stretching (TS), background mixing (BG), and impulse response convolution (IR). The best results are achieved when all augmentations are used.

| Augmentation | | | VLAFP on *FMA* | | |
|---|---|---|---|---|---|
| TS | BG | IR | Precision | Recall | F1 |
| No Augmentation | | | 26.54 | 24.62 | 25.54 |
| ✓ | | | 51.68 | 39.87 | 45.01 |
| | ✓ | | 66.16 | 60.70 | 63.31 |
| | | ✓ | 76.27 | 66.17 | 70.86 |
| ✓ | ✓ | | 67.95 | 63.45 | 65.62 |
| ✓ | | ✓ | 75.06 | 65.60 | 70.02 |
| | ✓ | ✓ | 76.84 | 66.93 | 71.54 |
| ✓ | ✓ | ✓ | **77.59** | **68.22** | **72.60** |

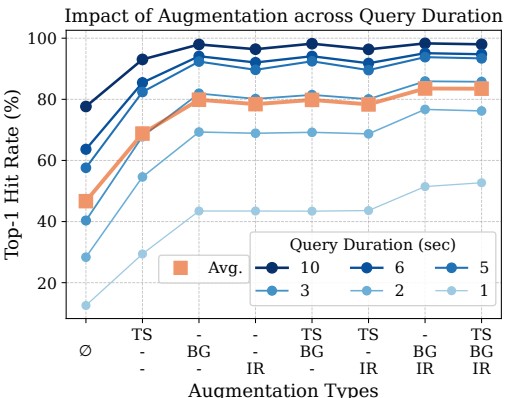

Figure 7: DTR results of VLAFP on FMA with different augmentation configurations.

Table 11: CBR results of VLAFP on FMA when trained with different numbers of positive samples #pos $\in \{1, 2, 3, 4, 5\}$ per anchor. A smaller number of positive samples yield better result.

| #pos | VLAFP on *FMA* | | |
|---|---|---|---|
| | Precision | Recall | F1 |
| 1 | **76.38** | 66.46 | 71.07 |
| 2 | 74.93 | **67.63** | **71.09** |
| 3 | **76.38** | 64.56 | 69.97 |
| 4 | 75.41 | 65.49 | 70.10 |
| 5 | 74.55 | 63.77 | 68.74 |

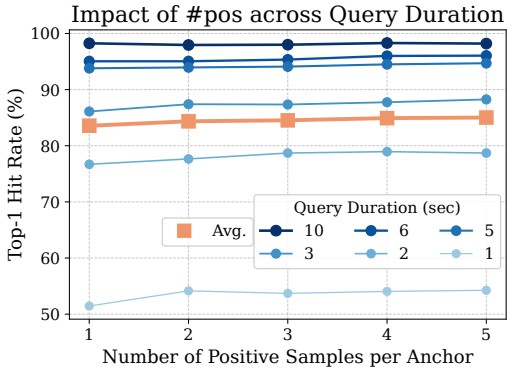

Figure 8: DTR results of VLAFP on FMA with different number of positive samples per anchor.

is achieved with 2 positive samples per anchor, while the precision is the highest with 1 and 3, as shown in Table 11. This indicates that a small number of positives yields competitive performance. Conversely, for the DTR task, increasing #pos from 1 to 2 improves the Top-1 Hit Rate significantly. Further increases provide little additional gain, as shown in Fig. 8.

***Impact of Other Hyperparameters.*** Finally, we evaluate the effectiveness of VLAFP with respect to other hyperparameters, including the number of blocks $L$, number of hidden dimensions $d$, number of heads $H$, and learning rate $\eta$. Detailed discussion and conclusion are included in Appendix C.

## 6 CONCLUSION

This paper proposes a novel Variable-Length Audio FingerPrinting method (VLAFP), which addresses the limitations of existing audio fingerprinting methods constrained by fixed-length segmentation. The variable-length capacity of VLAFP comes from a novel transformer-based architecture that consists of self-attention layers, which capture the inter-frame relationships, and cross-attention layers, which model the frame-to-segment relationships. Experiments on the commercial-broadcast retrieval and dummy-target retrieval show that our VLAFP outperforms existing state-of-the-arts. Future directions include making various parameters learnable (e.g., the segmentation threshold $\theta$), integrating semantics-based segmentation, and finetuning for variable-length fingerprinting. In addition, our proposed VLAFP can serve as a fundamental basis extended for many more tasks such as audio classification, speaker identification, keyword spotting, among others.

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

Table 12: A summary of notations.

| Symbol | Meaning | Symbol | Meaning | Symbol | Meaning |
|---|---|---|---|---|---|
| $a$ | audio signal | $n$ | audio length | $\mathbf{z}$ | audio fingerprint |
| $\mathbf{A}$ | audio in time-frequency rep. | $T$ | audio frame length | $F$ | number of frequency bins |
| $d$ | feature dimension | $d_h$ | head dimension | $H$ | number of attention heads |
| $L$ | number of blocks | $h^0$ | initial embedding | $h^l, \tilde{h}^l$ | frame-level embedding |
| $x_q^l$ | query variable | $x_{kv}^l$ | key/value variable | $s^l$ | seg.-level embedding |
| $a$ | anchor sample | $a^+$ | positive sample to $a$ | $P(a)$ | set of positive samples |
| $\mathcal{B}$ | batch size | $\tau$ | temperature parameter | $\eta$ | learning rate |
| $W_{stft}$ | STFT window size | $L_{frame}$ | hop length | $f_s$ | sampling rate |
| ; | concatenation | $\odot$ | Hadamard product | | |

## A DETAILS OF PRELIMINARIES

We summarize our used notations in Table 12. Additionally, we define four key components in VLAFP including normalization layers $\mathrm{RMSNorm}\,(\cdot)$, self-attention layers $\mathrm{SelfAttn}\,(\cdot)$, cross-attention layers $\mathrm{CrossAttn}\,(\cdot)$, and feedforward neural networks $\mathrm{FFN}\,(\cdot)$ (Vaswani et al., 2017; Zhang & Sennrich, 2019; Chen et al., 2021). For convenience, we let $x \in \mathbb{R}^{* \times d}$ denote a tensor of arbitrary shape with the last dimension equal to $d$.

***Root Mean Square Normalization Layers.*** Root mean square normalization layers are defined as $\mathrm{RMSNorm}\,(x) = x / \sqrt{\frac{1}{d} \sum_{i=1}^{d} x_i^2 + \epsilon}$, with a small constant $\epsilon$ for numerical stability. The normalization occurs across $d$ dimensions.

***Multi-Head Self-Attention Layers.*** Multi-head self-attention layers are denoted by $\mathrm{SelfAttn}\,(\cdot)$. Let $H$ denote the number of heads, then for each head $i$, the query, key, and value variables are computed as $(Q_i, K_i, V_i) = \left(xW_Q^i,\ xW_K^i,\ xW_V^i\right)$ with weights $W_Q^i, W_K^i, W_V^i \in \mathbb{R}^{d \times d_h}$, where $d_h$ is the feature dimension per head. Each head independently computes an intermediate representation, $x_i = \mathrm{softmax}\left(Q_i K_i^\mathsf{T} / \sqrt{d_h}\right) \cdot V_i \in \mathbb{R}^{* \times d_h}$, $i = 1, 2, \ldots, H$. All heads are concatenated and projected back to $d$ features through a projection layer with weights $W_O \in \mathbb{R}^{(H \cdot d_h) \times d}$,

$$\mathrm{SelfAttn}\,(x) = [x_1; x_2; \cdots; x_H] \cdot W_O \in \mathbb{R}^{* \times d} \tag{8}$$

***Multi-Head Cross-Attention Layers (Enhanced).*** Multi-head cross-attention Layers are denoted by $\mathrm{CrossAttn}\,(\cdot)$, are similar to self-attention layers but differ in the sources of the query, key, and value variables. Let $x_q \in \mathbb{R}^{H \times d}$ denote the source of the query, and $x_{kv} \times \mathbb{R}^{T \times d}$ denote the source of the key and value, we first apply RMS normalization, as $(\tilde{x}_q, \tilde{x}_{kv}) = (\mathrm{RMSNorm}\,(x_q), \mathrm{RMSNorm}\,(x_{kv}))$. As in multi-head self-attention layers, we assume $H$ heads. For each head $i$, the query, key, and value variables are computed as $(Q_i, K_i, V_i) = \left(\tilde{x}_q W_Q^i,\ \tilde{x}_{kv} W_K^i,\ \tilde{x}_{kv} W_V^i\right)$. Each head is computed independently: $x_i = \mathrm{softmax}\left(Q_i K_i^\mathsf{T} / \sqrt{d_h}\right) \cdot V_i \in \mathbb{R}^{H \times d_h}$, $i = 1, 2, \ldots, H$. All heads are concatenated and projected with $W_O \in \mathbb{R}^{(H \cdot d_h) \times d}$, in combination with a residual connection from $x_q$,

$$\mathrm{CrossAttn}\,(x_q, x_{kv}) = x_q + [x_1; \ldots; x_H] \cdot W_O \in \mathbb{R}^{H \times d} \tag{9}$$

Hence, we enhance the original cross-attention layers with $\mathrm{RMSNorm}$ and residual connections.

***Feedforward Neural Networks.*** Feedforward neural networks are defined as $\mathrm{FFN}\,(x) = (\mathrm{SiLU}\,(xW_1) \odot xW_3)\,W_2$, where $\mathrm{SiLU}\,(x) = x \cdot \sigma\,(x)$ is the sigmoid linear unit function (Elfwing et al., 2018). $W_1, W_3 \in \mathbb{R}^{d \times m}$ and $W_2 \in \mathbb{R}^{m \times d}$ are projection matrices where $m$ is $\left\lceil \alpha \cdot \frac{2}{3} \cdot 4d \right\rceil$ with a selected scaling factor $\alpha$.

## B DETAILS OF CONFIGURATION

### B.1 DATASETS & BASELINES

Table 13 summarizes the dataset statistics, including the number of audio samples, minimum, median, average, and maximum durations, total duration, and the type of each of the three datasets

Table 13: A summary of dataset statistics.

| Dataset | Split | #Audio | Duration (seconds / hours) | | | | | Type |
|---------|-------|--------|------|------|------|------|-------|------|
| | | | Min. | Med. | Avg. | Max. | Total | |
| FMA | Train | 10,000 | 6.2 | 30.0 | 30.0 | 30.0 | 83.3 h | Music |
| | Test | 500 | 30.0 | 30.0 | 30.0 | 30.0 | 4.2 h | |
| LibriSpeech | Train | 5,459 | 1.4 | 13.8 | 12.3 | 17.3 | 25.9 h | Speech |
| | Test | 2,620 | 1.3 | 5.8 | 7.4 | 35.0 | 5.4 h | |
| AudioSet | Train | 10,000 | 1.0 | 9.9 | 10.0 | 10.0 | 27.5 h | General |
| | Test | 500 | 3.9 | 9.9 | 10.0 | 10.0 | 1.4 h | |

used. Notably, our datasets cover various domains, including music, speech, and general audios. All audio files are converted to mono with a sampling rate of $8$ kHz in waveform format.

We compare VLAFP with the following five baselines. The first two are deep audio fingerprinting methods. The rest are general audio representation methods.

1. *NAFP* employs a CNN-based encoder to fingerprint fixed-length audio segments and is trained with the InfoNCE contrastive loss (Chang et al., 2021).

2. *AMG* uses a two-stage embedding approach. The first stage encodes audio, and the second stage feeds the embeddings through a Transformer-based encoder trained with a class-level loss function called Proxy-anchor Aligned Margin loss (PAM-Loss) (Su et al., 2024).

3. *wav2vec2* encodes audio with CNN and Transformer and is trained with the InfoNCE loss (Baevski et al., 2020). We use the *facebook/wav2vec2-base-960h* model version on Hugging Face.

4. *HuBERT* also encodes audio with CNN and Transformer. It predicts cluster assuagement for audio (Hsu et al., 2021). We use the *facebook/hubert-large-ls960-ft* model version on Hugging Face.

5. *AST* adapts the Vision Transformer (ViT) architecture to audio data (Gong et al., 2021). We use the *MIT/ast-finetuned-audioset-10-10-0.4593* model version on Hugging Face.

All baselines support only fixed-length audio processing. Hence, we apply fixed-length segmentation using a $1$-second window with a $0.5$-second hop to all of them. For NAFP and AMG, we use the GitHub code provided by the model authors, while for wav2vec2, HuBERT, and AST, we use the pretrained models available on Hugging Face. To our knowledge, there is no prior fine-tuning methodology specifically for the audio fingerprinting task. As fine-tuning these general audio models is beyond the scope of our work, we evaluated them as released.

We observe that HuBERT has extremely low precision and high recall. This can be due to (1) Different training objectives: HuBERT uses masked prediction leveraging contextual information without distortions, while our VLAFP is trained to minimize the distance between an audio segment and its distortions. (2) Different embedding characteristics: HuBERT's high-dimensional embeddings may retain more irrelevant information, while VLAFP produces compact, fingerprinting-specific representations. (3) Different temporal granularity: HuBERT may average out temporal details within its context windows, while VLAFP's frame-to-segment design preserves temporal information crucial for fingerprinting.

## B.2 VARIABLE-LENGTH SEGMENTATION

In light of addressing the multiple aforementioned drawback of fixed-length segmentation in Sec. 1, we propose a variable-length segmentation based on the spectral entropy of audio frames. VLAFP theoretically handles segments of any arbitrary length, however, handling extremely short (e.g., only 1 audio samples) or extremely long (e.g., 1 day long) segments is ineffective and impractical. Hence, we set a minimum length ($T_{\min} = 0.5$ seconds) and a maximum length ($T_{\max} = 5$ seconds) for the resulting segment lengths. In general, $T_{\min}$ is an initialized frame number to form a segment. So for a very short event, it will be combined with other frames to form a short segment (for example, of length $T_{\min}$). Since $T_{\min}$ is small, it won't significantly reduce fingerprint specificity. Based on

---

**Algorithm 2:** Spectral-Entropy Based Variable-Length Audio Segmentation

---

**1 Input:** audio signal $a = a[n]$, sampling rate $f_s$ ;

**2 Hyperparameters:** minimum segment length $T_{\min}$, maximum segment length $T_{\max}$, STFT window $W_{stft}$, frame length $L_{frame}$, z-score threshold $\theta$ ;

**3** Initialize empty set of segments $S = \{\}$ ;

**4 while** $a \neq \emptyset$ **do**

**5**      Initialize empty segment $s = \emptyset, T_s = 0$ ;

**6**      **while** $T_s < T_{\min}$ **and** $a \neq \emptyset$ **do**

**7**          Pop $L_{frame}$ samples from $a$ ;

**8**          Compute STFT frame $\mathbf{A}_i$ with window $W_{stft}$ ;

**9**          Add $\mathbf{A}_i$ to $s$ ;

**10**         Update $T_s \leftarrow T_s + L_{\text{frame}}/f_s$;

**11**      **end**

**12**      Calculate mean $\mu$ and std $\sigma$ of spectral entropy in $s$ ;

**13**      **while** $T_s < T_{\max}$ **and** $a \neq \emptyset$ **do**

**14**          Pop $L_{frame}$ samples from $a$ ;

**15**          Compute STFT frame $\mathbf{A}_j$ ;

**16**          Compute z-score $z_j$ using $\mu$ and $\sigma$ ;

**17**          **if** $z_j \leq \theta$ **then**

**18**             Add $\mathbf{A}_j$ to $s$ ;

**19**             Update $\mu$ and $\sigma$ ;

**20**             Update $T_s \leftarrow T_s + L_{\text{frame}}/f_s$;

**21**          **end**

**22**          **else**

**23**             **break** ;

**24**          **end**

**25**      **end**

**26**      Add $s$ to $S$ ;

**27 end**

**28 Output:** segment set $S$ with variable-length segments ;

---

our statistics, these short segments account for less than $1.7\%$ of the total audio segments and do not affect the detection of significant presence of a commercial or target.

Given an audio signal, we maintain a window representing the segment currently being formed. First, enough samples are added to meet the minimum length requirement of $T_{\min}$ seconds. Once satisfied, we compute the spectral entropy of the audio frames in the window. Audio frames are derived from STFT windows of size $W_{stft} = 1,024$ samples (128 ms) with a hop size of $L_{frame} = 256$ samples (32 ms), corresponding to a $75\%$ overlap.

We assume the spectral entropy within the window follows a normal distribution, with mean $\mu$ and standard deviation $\sigma$. To decide whether to include subsequent audio samples in the current window, we compute the z-score of spectral entropy of each new frame of $L_{frame}$ samples. The z-score is computed in relation to $\mu$ and $\sigma$. If the z-score is small than a threshold $\theta$ (e.g., $\theta = 2$), meaning the audio samples in consideration are close to those in the current window in terms of the spectral entropy, we expand the window to include these samples and update $\mu$ and $\sigma$ in the elongated window. Window expansion continues until either the window hits the maximum length or the z-score of the next audio frame is too greater than the threshold $\theta$, at which point a segment is formed. This process repeats until all audio samples are segmented, resulting in variable-length segments. The variable-length segmentation process is described in Algorithm 2.

Additionally, we propose three variable-length segmentation variants: (1) *No Silence* removes any silence (considered as 60 dB below the peak level) before applying the baseline segmentation. (2) *Pelt* is a change point detection algorithm and segments by detecting change point in spectral entropy values (Killick et al., 2012). (3) *Waveform* directly applies the baseline to the waveform entropy instead of the spectral entropy, we set the entropy threshold as $4$ in our CBR task. Their algorithms are described in Algorithm 3, Algorithm 4, and Algorithm 5, respectively.

**Algorithm 3:** Spectral-Entropy Based Variable-Length Audio Segmentation without Silence

1 **Input:** audio signal $a = a[n]$, sampling rate $f_s$;
2 **Hyperparameters:** minimum segment length $T_{\min}$, maximum segment length $T_{\max}$, STFT
   window $W_{stft}$, frame length $L_{frame}$, z-score threshold $\theta$ ;
3 Initialize empty set of segments $S = \{\}$ ;
4 **while** $a \neq \emptyset$ **do**
5      Initialize empty segment $s = \emptyset$, $T_s = 0$ ;
6      **while** $T_s < T_{\min}$ **and** $a \neq \emptyset$ **do**
7          Pop $L_{frame}$ samples from $a$;
8          **if** $L_{frame}$ *samples are silent* **then**
9              Continue;
10          **end**
11          **else**
12              Compute STFT frame $\mathbf{A}_i$ with window $W_{stft}$ ;
13              Add $\mathbf{A}_i$ to $s$ ;
14              Update $T_s \leftarrow T_s + L_{\mathrm{frame}}/f_s$;
15          **end**
16      **end**
17      Calculate mean $\mu$ and std $\sigma$ of spectral entropy in $s$ ;
18      **while** $T_s < T_{\max}$ **and** $a \neq \emptyset$ **do**
19          Pop $L_{frame}$ samples from $a$ ;
20          Compute STFT frame $\mathbf{A}_j$ ;
21          Compute z-score $z_j$ using $\mu$ and $\sigma$ ;
22          **if** $z_j \leq \theta$ **then**
23              Add $\mathbf{A}_j$ to $s$ ;
24              Update $\mu$ and $\sigma$ ;
25              Update $T_s \leftarrow T_s + L_{\mathrm{frame}}/f_s$;
26          **end**
27          **else**
28              **break** ;
29          **end**
30      **end**
31      Add $s$ to $S$ ;
32 **end**
33 **Output:** segment set $S$ with variable-length segments ;

---

**Algorithm 4:** Pelt Variable-Length Audio Segmentation

1 **Input:** audio signal $a = a[n]$;
2 **Hyperparameters:** minimum segment length $T_{\min}$, Jump $N_k$;
3 Initialize a Pelt model with $T_{\min}$ and $N_k$ ;
4 Fit Pelt model on $a$ ;
5 Predict segment set $S$ with the Pelt model ;
6 **Output:** segment set $S$ with variable-length segments ;

---

As shown in Fig. 6, *Waveform* segmentation has marginally better performance than the spectral entropy segmentation. However, it directly operates on the significantly larger volume of waveform values. For example, waveform segmentation needs to consider 8,000 values for a one-second audio of sampling rate 8kHz. This is over $25\times$ more than the spectral entropy segmentation, which considers about 30 resulting entropy values. Hence, we use the spectral entropy segmentation as our main segmentation method.

---

**Algorithm 5:** Waveform Variable-Length Audio Segmentation

---

1 **Input:** audio signal $a = a[n]$, sampling rate $f_s$;
2 **Hyperparameters:** minimum segment length $T_{\min}$, maximum segment length $T_{\max}$, STFT window $W_{stft}$, frame length $L_{frame}$ z-score threshold $\theta$ ;
3 Initialize empty set of segments $S = \{\}$ ;
4 **while** $a \neq \emptyset$ **do**
5     Initialize empty segment $s = \emptyset, T_s = 0$ ;
6     **while** $T_s < T_{\min}$ **and** $a \neq \emptyset$ **do**
7        Pop $L_{frame}$ samples from $a$;
8        Add $L_{frame}$ to $s$ ;
9        Update $T_s \leftarrow T_s + L_{\text{frame}}/f_s$;
10     **end**
11     Calculate mean $\mu$ and std $\sigma$ of spectral entropy in $s$ ;
12     **while** $T_s < T_{\max}$ **and** $a \neq \emptyset$ **do**
13        Pop $L_{frame}$ samples from $a$ ;
14        Compute z-score $z_j$ using $\mu$ and $\sigma$ ;
15        **if** $z_j \leq \theta$ **then**
16           Add $L_{frame}$ to $s$ ;
17           Update $\mu$ and $\sigma$ ;
18           Update $T_s \leftarrow T_s + L_{\text{frame}}/f_s$;
19        **end**
20        **else**
21           **break** ;
22        **end**
23     **end**
24     Apply STFT to $s$ with $W_{stft}$ and add to $S$ ;
25 **end**
26 **Output:** segment set $S$ with variable-length segments ;

---

### B.3 AUDIO AUGMENTATION AND TIME-FREQUENCY REPRESENTATION

#### B.3.1 AUDIO AUGMENTATION

To create positive samples in contrastive learning, we distort each segment with a chain of time-stretching, background noise mixing, impulse response convolution. For a given audio segment, we randomly sample a time-scaling factor from $[0.8, 1.2]$ to apply a time-stretch transformation, which either slows down or speeds up the audio depending on whether the sampled value is less than or greater than 1. Next, we randomly select a background noise excerpt from a pool of $2,142$ candidates and mix it with the segment. Lastly, we apply impulse response convolution with an impulse response signal randomly selected from a pool of $345$ candidates. All augmentations are performed on audio samples in the time-domain representation.

#### B.3.2 TIME-FREQUENCY REPRESENTATION

For resulting segments and their distortions, we apply a mel-spectrogram transformation with $F = 256$ mel bands, an STFT window of $W_{stft} = 1024$ samples, and a hop length of $L_{frame} = 256$ samples. Given the dataset sampling rate of $f_s = 8000$ Hz, the STFT window corresponds to 128 milliseconds and the hop length corresponds to 32 milliseconds, meaning that each audio frame is derived from 128 milliseconds with a $75\%$ overlap between adjacent frames. Additionally, we apply filtering with a minimum frequency as 300 Hz, a maximum frequency of 4000 Hz, a dynamic range of 80 dB, and a signal-to-noise ratio of $[1, 10]$.

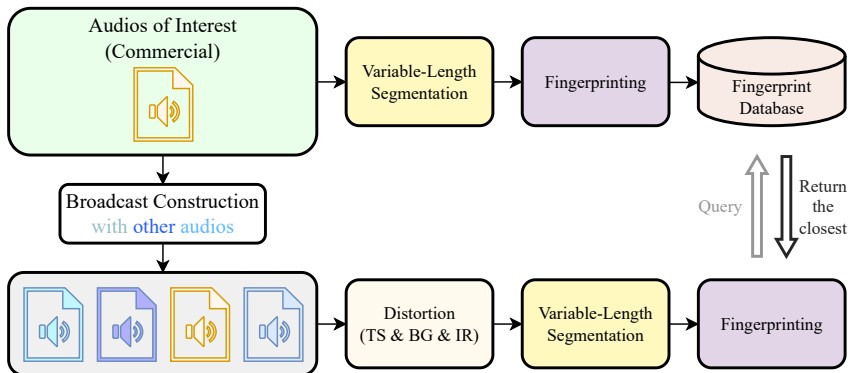

Figure 9: A workflow diagram of the Commercial-Broadcast Retrieval (CBR).

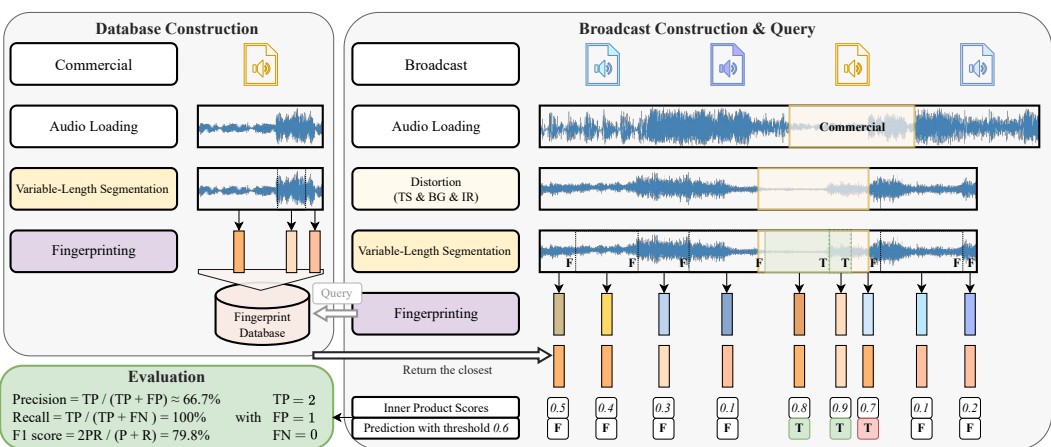

Figure 10: Details of database construction and broadcast construction & query in Commercial-Broadcast Retrieval (CBR).

### B.4 TASK CONFIGURATIONS

#### B.4.1 COMMERCIAL-BROADCAST RETRIEVAL (CBR)

As shown in Fig. 9, a commercial is segmented and fingerprinted to construct a fingerprint database. We then simulate the broadcast by randomly selecting 19 additional audios from the test set, concatenating them with the commercial in a shuffled order, and applying audio distortions including time-stretch transformation, background noise mixing, and impulse response convolution. The simulated stream is subsequently segmented and fingerprinted, and similarity is measured by the inner product score to retrieve the most similar fingerprint from the database. A query segment is considered correctly identified as the commercial if its retrieved fingerprint has a high inner product score relative to a threshold. Fig. 10 provides an example on how these scores are used with a threshold, along with other details. Since the optimal score threshold varies across fingerprinting methods, we select the threshold that yields the best F1 score for each method to ensure a fair comparison. We define true positive (TP) as the number of queried segments that are correctly identified as the commercial, false positive (FP) as the number of segments wrongly identified as the commercial, and false negative (FN) as the number of segments wrongly identified as unrelated. Accordingly, our metrics are defined as $\text{Precision} = \frac{\text{TP}}{\text{TP+FP}}$, $\text{Recall} = \frac{\text{TP}}{\text{TP+FN}}$, and $\text{F1} = \frac{2 \cdot \text{Precision} \cdot \text{Recall}}{\text{Precision+Recall}}$.

#### B.4.2 DUMMY-TARGET RETRIEVAL (DTR)

As shown in Fig. 11, DTR constructs a database on both unrelated audios (dummy) and audios of interest (target). Each query corresponds to a distorted version of a target audio. As depicted in

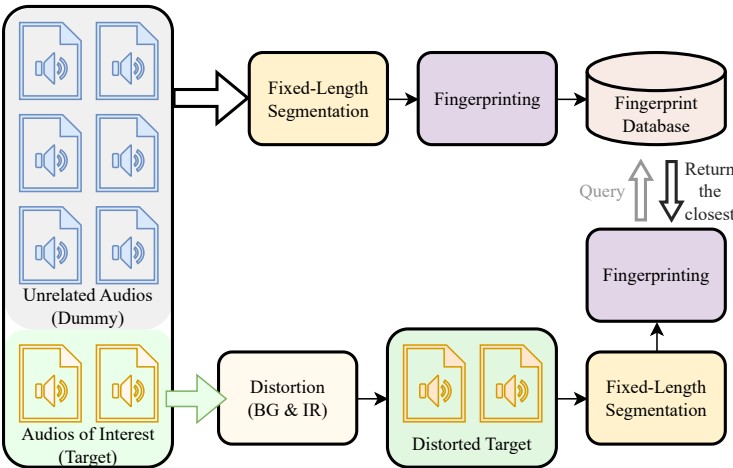

Figure 11: A workflow diagram of the Dummy-Target Retrieval (DTR).

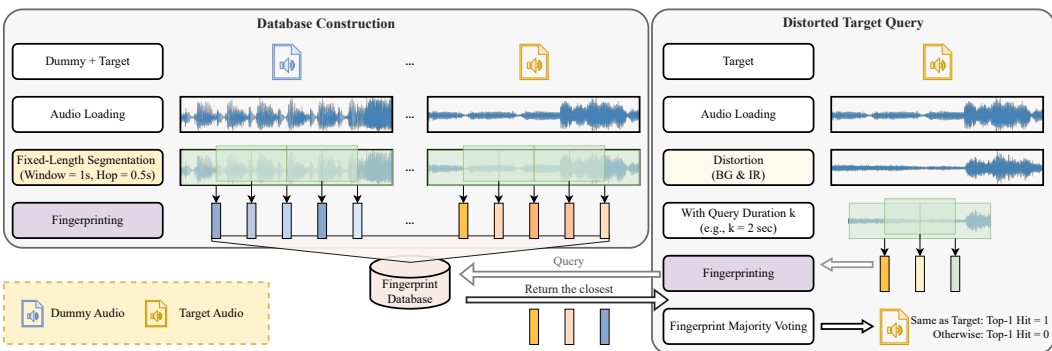

Figure 12: Details of database construction and query in Dummy-Target Retrieval (DTR).

Fig. 12, DTR construct a database by getting all segments from both dummy and target audios. In the query stage, for a query of duration $k$ seconds, DTR fingerprints $2k - 1$ segments (using a 1-second window with a 0.5-second hop) and retrieves $2k - 1$ segments from the database. Since these retrieved segments may come from different audios, we identify the audio that contributes the majority of retrieved segments, and designate that as the retrieved audio. For each query, the Top-1 Hit is 1 if the correct audio is retrieved and 0 otherwise. For this task, we adopt fixed-length segmentation (1 sec) for VLAFP for two reasons: (1) To illustrate the effectiveness of our proposed transformer-based architecture, especially compared to NAFP (Chang et al., 2021), the only difference is the model architecture while all other settings are the same in terms of segmentation, batch size, number of positive samples, etc. (2) Since the audios of interests (target) are fingerprinted based on 1-sec segments, it is not in favor of training with variable-length. In addition, since time-stretching is not present in the target audio distortion, we do not apply time-stretching for audio augmentation. As a result, we created a database containing $611,422$ segments ($581,922$ for dummy and $29,500$ for target) from the *FMA* dataset.

## B.5 TRAINING CONFIGURATIONS

Unless otherwise specified, we experiment with the following parameters. We select the same number for all hidden feature dimensions $d = d_1 = d_2 = 256$ and number of blocks $L = 4$. The number of heads in both self-attention layers and cross-attention layers is set as $H = 8$ where each head has a dimension of 256, the scaling factor in the feedforward neural networks is $\alpha = 32$.

We train VLAFP for 100 epochs with the Adam optimizer at a learning rate $\eta$ of $10^{-5}$. For the ablation study and impacts of hyperparameters, we save compute by training only for 10 epochs,

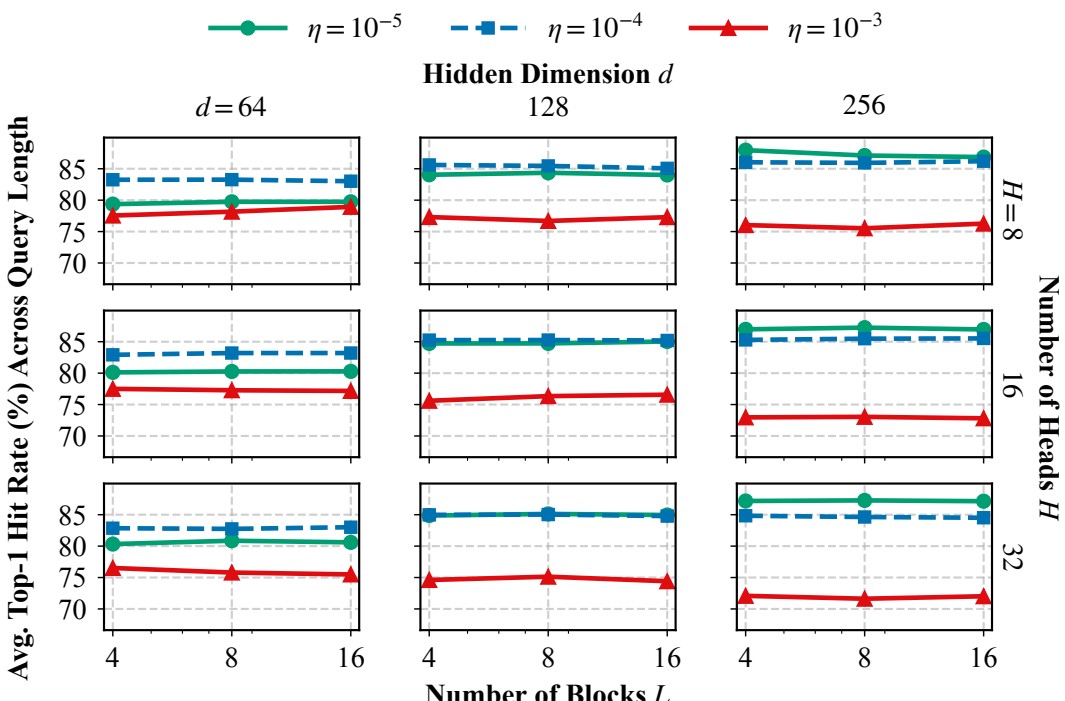

Figure 13: A sensitivity study of hyperparameter sensitivity of VLAFP. Each row is for a specific number of heads $H \in \{8, 16, 32\}$ and each column is for a specific hidden dimension $d \in \{64, 128, 256\}$. In each figure, each line is for a specific learning rate $\eta \in \{10^{-5}, 10^{-4}, 10^{-3}\}$, and each dot is for a specific number of blocks $L \in \{4, 8, 16\}$. The average accuracy across query duration is reported. See Sec. C for detailed discussions.

since our focus is the relative performance in different settings. For each segment, we create three positive samples. In Eq. 7, the temperature parameter $\tau = 0.05$ and the batch size $|\mathcal{B}| = 60$. For all baselines, we adopt the default parameters either from their officially released code or, if unavailable, as specified in their respective papers. All training is performed on machines with 16 CPUs, 128 GB of memory, and an NVIDIA L4 GPU.

## C IMPACT OF ADDITIONAL HYPERPARAMETERS

We evaluate the effectiveness of VLAFP under a broad range of hyperparameters, including number of blocks $L \in [4, 8, 16]$, number of hidden dimensions $d \in [64, 128, 256]$, number of heads $H \in [8, 16, 32]$, and learning rate $\eta \in [10^{-5}, 10^{-4}, 10^{-3}]$. This results in $81 = 3 \times 3 \times 3 \times 3$ model instances, each trained and evaluated on the DTR task described in Sec. 5.2. The average top-1 hit rate across $\{1, 2, 3, 5, 6, 10\}$ seconds is reported in Fig. 13. From Fig. 13, we observe: (1) *Learning rate $\eta$:* The best performance consistently comes from smaller learning rates, either from $\eta = 10^{-5}$ (blue circles) or $\eta = 10^{-4}$ (orange squares), while $\eta = 10^{-3}$ (green triangles) performs significantly worse across settings. (2) *Number of blocks $L$:* Within each plot, the curves are relatively flat across $L = 4, 8, 16$, indicating that $L$ has little influence on performance. (3) *Hidden dimension $d$:* Across each row, increasing $d$ from 64 to 256 improves performance under smaller learning rates ($10^{-5}$ and $10^{-4}$) but degrades performance when $\eta = 10^{-3}$. This suggests that larger $d$ is beneficial with appropriately small learning rates. (4) *Number of heads $H$:* For each column, as $H$ increases from 8 to 32, performance at learning rate $\eta = 10^{-3}$ and $\eta = 10^{-4}$ deteriorates, while model with $\eta = 10^{-5}$ remains stable. Based on the result, we select $d = 256$, $L = 4$, $H = 8$, and $\eta = 10^{-5}$ as the default configuration.

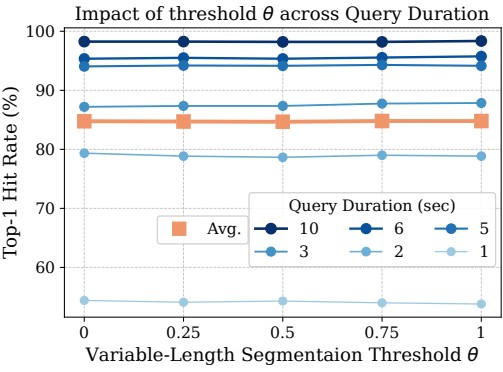

Table 14: CBR results of VLAFP on FMA when trained with different segmentation threshold $\theta \in \{0, 0.25, 0.50, 0.75, 1, 2\}$. Higher values indicate better performance.

| Segmentation | VLAFP on *FMA* | | |
|---|---|---|---|
| Threshold $\theta$ | Precision | Recall | F1 |
| 0 | 79.28 | **71.63** | 75.26 |
| 0.25 | **82.02** | 70.42 | 75.78 |
| 0.50 | 80.73 | 70.89 | 75.49 |
| 0.75 | 80.62 | 71.52 | **75.80** |
| 1 | 81.00 | 70.15 | 75.19 |
| 2 | 74.86 | 65.24 | 69.72 |

Figure 14: DTR results of VLAFP on FMA with $\theta \in \{0, 0.25, 0.50, 0.75, 1\}$.

Table 15: CBR results of VLAFP on FMA with YAMNet.

Table 16: DTR results of VLAFP on FMA when trained with different segmentation methods.

| Segmentation | VLAFP on *FMA* | | |
|---|---|---|---|
| | Precision | Recall | F1 |
| main (Ours) | **81.00** | 70.15 | 75.19 |
| YAMNet | 75.76 | **77.40** | **76.57** |

| Segmentation | Number of Seconds in Query on FMA (Top-1 Hit Rate) | | | | | | |
|---|---|---|---|---|---|---|---|
| | 1 | 2 | 3 | 5 | 6 | 10 | Avg. |
| main (Ours) | **51.45** | **76.65** | **86.00** | **93.90** | **95.10** | **98.25** | **83.56** |
| YAMNet | 44.00 | 68.50 | 80.80 | 90.60 | 92.55 | 97.45 | 78.98 |

## D  ADDITIONAL EXPERIMENTS ON VARIOUS ENTROPY THRESHOLDS $\theta$

We further investigate the impact of the entropy threshold $\theta$ in a more refined range $\theta \in \{0, 0.25, 0.50, 0.75, 1\}$. As shown in Table 14 and Fig. 14, our VLAFP exhibits consistently strong performance across these values. Similar trends were observed for the DTR task.

## E  ADDITIONAL EXPERIMENTS ON SEMANTICS-BASED SEGMENTATION

We further investigate the possibility of using our VLAFP with semantics-based segmentation. We adopt YAMNet as a semantics-based segmentation approach, which classifies audio frames into 521 classes, and we select the most likely class per frame (yam). For example, if YAMNet analyzes an audio clip of six frames (each ∼1 second) and labels them as [speech, speech, singing, guitar, guitar, guitar], we split the audio into three segments: 0–2 seconds, 2–3 seconds, and 3–6 seconds. We compared YAMNet against our spectral entropy-based method on the CBR and DTR tasks. As observed in Table 15 and Table 16, YAMNet achieves lower precision and higher recall and F1 for the CBR task, while YAMNet consistently performs worse on the DTR task. This highlights the potential of future research to develop segmentation methods tailored to different tasks. Importantly, our VLAFP enables the use of such task-adaptive segmentation, since previous fixed-length methods are constraint to fixed-length segmentation.

## F  ADDITIONAL EXPERIMENTS ON KEYWORD SPOTTING

We evaluated VLAFP on the keyword spotting task using the *SpeechCommand* dataset in the *SU-PERB benchmark* (Yang et al., 2021). SpeechCommand is a classification dataset with ten keyword classes, a silence class, and an unknown class. More details can be found in (Warden, 2018).

Our VLAFP achieves $40.00\%$ accuracy, while general-purpose audio representation models like wav2vec2-base reported $96.23\%$ and HuBERT-large reported $95.29\%$ accuracy. This performance difference is expected, as VLAFP is specifically designed for audio fingerprinting rather than multi-

Table 17: CBR results on Dataset *BAF*. Values are reported as percentages (%) for *precision*, *recall*, and *F1-score*, with the best scores highlighted in **bold** and second best in *italics*.

| Method | Dataset BAF | | |
|---|---|---|---|
| | Precision | Recall | F1 |
| wav2vec2 | 12.52 | **58.29** | 20.62 |
| HuBERT | 14.17 | 38.60 | 20.73 |
| AST | 15.49 | 54.62 | 24.14 |
| AMG | 16.94 | 36.26 | 23.10 |
| NAFP | 39.62 | 27.59 | 32.52 |
| **VLAFP (Ours)** | **40.87** | 29.39 | **34.19** |

Table 18: CBR results on Dataset *BAF* with baselines WavLM (Chen et al., 2022) and Whisper (Radford et al., 2023). Values are reported as percentages (%) for *precision*, *recall*, and *F1-score*, with the best scores highlighted in **bold** and second best in *italics*.

| Method | Dataset BAF | | |
|---|---|---|---|
| | Precision | Recall | F1 |
| WavLM | 13.11 | **55.28** | 21.19 |
| Whisper | 13.70 | 42.83 | 20.76 |
| **VLAFP (Ours)** | **40.87** | 29.39 | **34.19** |

class classification tasks like keyword spotting. We recognize the potential of adapting VLAFP for broader audio tasks through classification-based objectives or fine-tuning strategies. Future research may further explore this direction.

# G ADDITIONAL EXPERIMENTS ON DATASET *BAF*

We conduct out-of-domain evaluation on a real world Broadcast Audio Fingerprinting (BAF) dataset (Cortès et al., 2022). Different from the synthetic approach, BAF records broadcasts from TV shows. In our experiments, we select commercials (named as *references* in BAF) that (1) are unanimously confirmed to appear in the broadcast by three annotators and (2) last fewer than 10 seconds in the broadcast, resulting 135 commercial-broadcast pairs. Since the dataset does not contain a training set, we leverage the trained VLAFP from the FMA dataset, making the evaluation out-of-domain. As shown in Table 17, VLAFP achieves the best precision and F1 scores among all methods. Note that these scores are obtained by trying different similarity thresholds, and we use the threshold that corresponds to the best F1. We observe again general audio representation baselines, such as wav2vec2 and AST, are inclined to identify broadcast segments as commercials, leading to lower precision and higher recall. When we loosen the threshold to increase VLAFP's recall to be 58.29%, VLAFP has a precision of 15.19%, still better than wav2vec2's 12.52%. This validates the effectiveness and superiority of VLAFP on data under real-world distortion.

Moreover, we compare with more recent self-supervised learning models, WavLM (Chen et al., 2022) and Whisper (Radford et al., 2023). We use *microsoft/wavlm-base* and *openai/whisper-base* from Hugging Face for these models. As shown in Table 18, they have similar results as other general audio representation baselines. Our VLAFP outperforms both models in terms of precision and F1 scores.

