# OpenReview forum: "Variable-Length Audio Fingerprinting"
_ICLR.cc/2026/Conference — Submitted to ICLR 2026_

### Official Review · Reviewer_483Z · 2025-10-29

**Soundness:** 3
**Presentation:** 4
**Contribution:** 4
**Rating:** 6
**Confidence:** 4

**Summary:**

The paper introduces a Transformer-based framework for the fingerprinting of variable-length audio fingerprinting. The approach in this paper dynamically segments audio based on spectral entropy and learns fingerprints through a dual-attention Transformer using supervised contrastive learning. Judging by the results reported in the paper, the approach successfully captures temporal and semantic coherence in variable-length signals, achieving state-of-the-art performance in both live audio identification and retrieval tasks on multiple datasets. Ablation experiments confirm the importance of the variable-length design and dual-attention structure for robust fingerprint generation under diverse distortions.

**Strengths:**

1. The approach proposed is reasonably novel, both in architecture and in the problem it solves.
2. The formulation of segmentation logic and the learning processes is theoretically sound and verifiable.
3. The work is clearly presented, and the appendices provide additional details that allows for better understanding (with one important caveat)
3. Robust fingerprinting of variable-length sequences (like audio), especially against resampling and speed adjustments, is of significant interest in signal processing-related tasks.

**Weaknesses:**

The segmentation threshold, being manually tuned rather than learned, might limit domain-transferability. The experiments on the effect of entropy thresholds also seem to be lacking in granularity.
The experiments reported seem to indicate
Presentation of Variable Length Segmentation algorithm is contained in Appendix B rather than the main body of the manuscript. As appendices may not be published in the proceedings, leaving out such an important part may not be desirable.

**Questions:**

1. The entropy threshold for segmentation seems empirically derived, but the exploration and discussion could be a bit more thorough. Do the authors plan on further investigating alternative methods of segmentation (such as more semantically-related approaches)?
2. The authors may consider moving some of the discussions regarding the Variable Length Segmentation algorithm to the main body of the text.

---

> ### Author Response · Authors · 2025-11-17
> **Initial Response**
>
> Thank you for the encouraging feedback!
> We have carefully addressed all your comments, and the revised version is now available in the uploaded PDF. Please feel free to let us know if you have any additional questions or suggestions—we'd be glad to discuss them.
>
> ---
>
> ### **Comment 1: discuss the learnability of the segmentation threshold**
> > The segmentation threshold, being manually tuned rather than learned, might limit domain-transferability.
>
> Thank you for raising this important point.
> In our current design, the segmentation threshold is treated as a data-driven hyperparameter, tuned for performance on the specific dataset.
> We agree that making this threshold learnable would enhance domain-transferability, and we see this as a promising direction for future research.
> Notably, such adaptability becomes viable only with variable-length approaches (our VLAFP).
> **We have now included this discussion in the future work section to highlight its potential.**
>
> ---
>
> ### **Comment 2: report on finer segmentation threshold granularity**
> > The experiments on the effect of entropy thresholds also seem to be lacking in granularity.
>
> Thank you for your insightful comment.
> As suggested, we have expanded our analysis by using a finer-grained set of segmentation thresholds, specifically $\theta$ = 0, 0.25, 0.50, 0.75, and 1.
> | $\theta$    | Precision | Recall    | F1        |
> | ---- | --------- | --------- | --------- |
> | 0    | 79.28     | **71.63** | 75.26     |
> | 0.25 | **82.02** | 70.42     | 75.78     |
> | 0.5  | 80.73     | 70.89     | 75.49     |
> | 0.75 | 80.62     | 71.52     | **75.80** |
> | 1    | 81.00     | 70.15     | 75.19     |
> | 2    | 74.86     | 65.24     | 69.72     |
>
> The table shows that our VLAFP exhibits consistently strong performance across these values.
> Similar trends were observed for the DTR task.
> **We have added results for both tasks to the paper.**
>
> ---
>
> ### **Comment 3: report results on semantics-based segmentation**
> > The entropy threshold for segmentation seems empirically derived, but the exploration and discussion could be a bit more thorough. Do the authors plan on further investigating alternative methods of segmentation (such as more semantically-related approaches)?
>
> Thank you for this insightful suggestion! We fully agree that exploring semantics-based segmentation is an exciting future direction.
> As recommended, we adopted YAMNet as a semantics-based segmentation approach.
> YAMNet classifies audio frames into 521 classes, and we select the most likely class per frame.
>
> For example, if YAMNet analyzes an audio clip of six frames (each ~1 second) and labels them as [speech, speech, singing, guitar, guitar, guitar], we split the audio into three segments: 0–2 seconds, 2–3 seconds, and 3–6 seconds.
> Such that each segment has the same label among its audio frames.
>
> We then compared YAMNet against our spectral entropy-based method on the CBR and DTR tasks.
>
> - CBR Task Performance
>
> | Segmentation Approach   | Precision | Recall | F1    |
> | ----------------------- | --------- | ------ | ----- |
> | Spectral Entropy (Ours) | **81.00**     | 70.15  | 75.19 |
> | YAMNet                  | 75.76     | **77.40**  | **76.57** |
>
> - DTR Task Performance
>
> |Top-1 Hit Rate | Query Length |||| | | |
> | ----------------------- | ------ | ------ | ------ | ------ | ------ | ------- | ------- |
> | Segmentation Approach  | 1 sec  | 2 sec  | 3 sec  | 5 sec  | 6 sec  | 10 sec | Average |
> | Spectral Entropy (Ours) | **51.45** | **76.65** | **86.00** | **93.90** | **95.10** | **98.25** | **83.56** |
> | YAMNet                  | 44.00  | 68.50  | 80.80  | 90.60  | 92.55  | 97.45   | 78.98   |
>
>
> As observed, YAMNet achieves lower precision and higher recall and F1 for the CBR task.
> YAMNet consistently performs worse on the DTR task.
> This highlights the potential of future research to develop segmentation methods tailored to different tasks. Importantly, **our VLAFP enables the use of such task-adaptive segmentation**, since previous fixed-length methods are constraint to fixed-length segmentation.
> **We have included the result in the paper.**
>
> ---
>
> ### **Comment 4: move content from appendix to the main body**
> > The experiments reported seem to indicate Presentation of Variable Length Segmentation algorithm is contained in Appendix B rather than the main body of the manuscript. As appendices may not be published in the proceedings, leaving out such an important part may not be desirable. The authors may consider moving some of the discussions regarding the Variable Length Segmentation algorithm to the main body of the text.
>
> Thank you for the valuable suggestion. Due to space constraints, we were unable to fully move the content from Appendix B.
>  However, to improve clarity while maintaining focus, **we have added a concise pseudocode description of the Variable-Length Segmentation algorithm to the main body and expanded the related discussion.**

---

> ### Comment · Area_Chair_grNQ · 2025-11-25
>
> Dear Reviewer 483Z,
>
> The authors have responded to your reviews. Please review and provide your feedback and responses.
>
> Best,
>
> Your AC

---

### Official Review · Reviewer_3yp7 · 2025-10-30

**Soundness:** 2
**Presentation:** 2
**Contribution:** 2
**Rating:** 2
**Confidence:** 4

**Summary:**

The paper proposes an attention-based audio fingerprinting method capable of handling variable-length input audio.

**Strengths:**

The core idea leverages two attention mechanisms—inter-frame and frame-to-segment attention—to capture both local and global contextual information from audio frames, which is a reasonable design choice.

**Weaknesses:**

However, several significant issues undermine the paper’s contribution and rigor:

1. **Lack of Technical Precision**: The mathematical formulation is often imprecise. For instance, in Section 3.1, the use of an approximation symbol (≈) is misleading; the intent appears to be minimizing the distance between *z* and *z*′ , which should be explicitly stated as an optimization objective. Moreover, symbols are frequently introduced without definition—e.g., *H* and *s* appear abruptly in the Frame-to-Segment Cross-Attention section, and *z* is used in the loss function (Section 3.3) without prior explanation.
2. **Limited Novelty**: The proposed architecture does not introduce substantial methodological innovation beyond combining existing attention mechanisms in a straightforward manner.
3. **Insufficient Experimental Detail**: Key implementation details are missing. For example, it is unclear how the baseline models (Wav2Vec2, HuBERT, and AST) were adapted for the fingerprinting task—specifics regarding classifiers, optimizers, training protocols, etc., are omitted. This omission raises concerns about the fairness of the reported comparisons and may suggest an attempt to understate baseline performance.
4. **Unsubstantiated Claims**: The introduction identifies three specific limitations of prior work—Loss of Natural Boundaries, Redundant or Noisy Context, and Distortion Incompatibility—but the paper never empirically investigates how these issues affect performance or how the proposed method mitigates them. The conclusion that “VLAFP likely benefits most from variable-length segmentation” is presented as intuitive rather than evidence-based, weakening the motivation and contribution of the work.
5. **Missing Methodological Clarification**: The computation of the spectral entropy score—a key evaluation metric—is not described, making it difficult to interpret or reproduce the results.

Overall, while the problem setting is relevant, the current presentation lacks the technical rigor, novelty, and empirical validation required for publication in its present form.

**Questions:**

The above weaknesses.

---

> ### Author Response · Authors · 2025-11-17
> **Initial Response**
>
> Thank you for your comments.
> We have addressed each of your points below and are happy to provide further clarification or additional information if needed.
>
> ---
>
> ### **Comment 1: describe with more precision**
> > In Section 3.1, the use of an approximation symbol (≈) is misleading; the intent appears to be minimizing the distance between z and z′ , which should be explicitly stated as an optimization objective.
> H and s appear abruptly in the Frame-to-Segment Cross-Attention section, and z is used in the loss function (Section 3.3) without prior explanation.
>
> We thank the reviewer for pointing this out.
> **We have revised the manuscript.**
> - The optimization objective is now stated as to learn the fingerprinting model $f = argmax_{f_\phi} E_a[f_\phi(a)^T_\phi(a')]$ in Section 3.1.
>
> We have also defined the following symbols in Section 3.2 and 3.3.
> - Let $H$ denote the number of segment embeddings in a block.
> - Let $s^{l}$ denote the segment embeddings in the $l$(th) block.
> - Let $\mathbf{z}=f(a)$ denote the fingerprint of an audio $a$.
>
> ---
>
> ### **Comment 2: explain the novelty in VLAFP's architecture**
> > The proposed architecture does not introduce substantial methodological innovation beyond combining existing attention mechanisms in a straightforward manner.
>
> We would like to clarify that the proposed VLAFP model introduces several novel contributions:
> 1. To the best of our knowledge, it is the first deep variable-length audio fingerprinting method.
> 2. It incorporates a novel variable-length segmentation approach.
> 3. It presents a task-oriented architecture consisting of five stages, which goes beyond merely combining attention mechanisms.
>
> In particular, the **Segment Embedding Initialization** and **Fingerprint Summarization** stages are specifically designed and adapted for the fingerprinting task, highlighting methodological contributions beyond standard attention-based designs.
>
> ---
>
> ### **Comment 3: describe baseline details**
> > It is unclear how the baseline models (Wav2vec2, HuBERT, and AST) were adapted for the fingerprinting task.
>
> Thank you for pointing this out.
> To ensure a fair comparison without cherry-picking, we used the model versions as provided on the Hugging Face model hub homepages:
> - wav2vec 2.0: _facebook/wav2vec2-base-960h_
> - HuBERT: _hubert-large-ls960-ft_
> - AST: _MIT/ast-finetuned-audioset-10-10-0.4593_
>
> We primarily follow the usage instructions of these baseline methods by using their implementations directly. However, we found the reviewer's comments insightful and are currently conducting additional experiments.
> **We have added these details to the paper for clarity.**
>
> ---
>
> ### **Comment 4: explain how VLAFP mitigates the three limitations**
> > The introduction identifies three specific limitations of prior work—Loss of Natural Boundaries, Redundant or Noisy Context, and Distortion Incompatibility—but the paper never empirically investigates how these issues affect performance or how the proposed method mitigates them. The conclusion that “VLAFP likely benefits most from variable-length segmentation” is presented as intuitive rather than evidence-based, weakening the motivation and contribution of the work.
>
> We would like to clarify a misinterpretation of our statement regarding variable-length segmentation. Our full sentence reads:
> > _**Intuitively, VLAFP likely benefits most from variable-length segmentation on LibriSpeech, which avoids forming segments that cross speech-silence boundaries and thus produces more coherent fingerprints.**_
>
> This observation is grounded in experimental evidence: VLAFP outperforms other baselines more substantially on LibriSpeech (8%+) than on FMA (5%) or AudioSet (1%), as shown in Table 2 - CBR results on FMA.
>
> Our results demonstrate that variable-length segmentation consistently improves accuracy over fixed-length baselines. Notably, it directly mitigates distortions such as time stretching, which are incompatible with fixed-length segmentation.
> This provides clear benefits supported by experimental evidence.
>
> ---
>
> ### **Comment 5: describe the concept of _spectral entropy_**
> > The computation of the spectral entropy score—a key evaluation metric—is not described, making it difficult to interpret or reproduce the results.
>
> _Spectral entropy_ is an uncertainty measure in the distribution of spectral energy within an audio frame. To compute it, we apply a short-time Fourier transform to the audio signal to obtain its frequency energy distribution, normalize these energy values to form a probability distribution, and then compute the Shannon entropy of this distribution. Intuitively, a low spectral entropy indicates that the energy is concentrated in a narrow frequency range, as is typical of pure tones, whereas high spectral entropy reflects a more uniform spectral distribution, such as in white noise.
> **We have expanded the explanation of spectral entropy in the main text, and we now include a reference for its computation details.**

---

> ### Comment · Area_Chair_grNQ · 2025-11-25
>
> Dear Reviewer 3yp7,
>
> The authors have responded to your reviews. Please review and provide your feedback and responses.
>
> Best,
>
> Your AC

---

### Official Review · Reviewer_Y1ti · 2025-10-30

**Soundness:** 4
**Presentation:** 4
**Contribution:** 3
**Rating:** 6
**Confidence:** 4

**Summary:**

This paper proposes Variable-Length Audio FingerPrinting (VLAFP), a novel deep learning framework for audio fingerprinting that overcomes the limitations of fixed-length segmentation prevalent in existing methods. VLAFP introduces a transformer-based architecture with dual-attention mechanisms—self-attention to model inter-frame dependencies and cross-attention to aggregate frame-level features into segment-level embeddings. A key innovation is the variable-length segmentation strategy based on spectral entropy, which dynamically determines segment boundaries to preserve semantic coherence. The method is evaluated on two tasks—Commercial-Broadcast Retrieval (CBR) and Dummy-Target Retrieval (DTR)—across three real-world datasets (FMA, LibriSpeech, AudioSet), demonstrating superior performance over state-of-the-art baselines in terms of precision, recall, F1, and top-1 hit rate. The paper also includes comprehensive ablation studies and hyperparameter analysis, supporting the effectiveness of the proposed design choices.

**Strengths:**

1. Novelty of Variable-Length Processing: To the best of the reviewers’ knowledge, VLAFP is the first deep audio fingerprinting model capable of handling variable-length inputs during both training and inference. This addresses a critical limitation in existing methods and enables more natural and semantically meaningful segmentation.

2. Well-Motivated and Effective Segmentation Strategy: The spectral entropy-based segmentation is well-justified and grounded in signal processing principles. The method adaptively forms segments based on acoustic homogeneity, avoiding unnatural cuts across silence or phonetic boundaries, which is particularly beneficial for speech and music signals.

3. Strong and Consistent Empirical Results: VLAFP consistently outperforms both specialized audio fingerprinting models (NAFP, AMG) and general-purpose audio representation models (wav2vec2, HuBERT, AST) across multiple datasets and evaluation metrics. The improvements are significant, especially in the CBR task, with F1 scores increasing by over 5 points on FMA and LibriSpeech.

**Weaknesses:**

1. Computational Overhead at Inference: The paper notes that VLAFP has a longer inference time due to the overhead of handling variable-length segments. This could be a practical limitation for real-time applications, and the trade-off between accuracy and efficiency is not deeply analyzed.

2. Evaluation on Synthetic Distortions: The audio augmentations (time-stretching, background noise, impulse response) are synthetically applied. While standard in the field, real-world broadcast distortions may be more complex and heterogeneous, and the robustness of VLAFP in truly uncontrolled environments remains to be validated.

**Questions:**

1. The paper compares against wav2vec 2.0 and HuBERT but does not specify which model variants (e.g., Base, Large) were used. Could the authors clarify the exact versions and whether they were fine-tuned for the fingerprinting task? Additionally, would more recent self-supervised learning (SSL) models—such as WavLM or Whisper—yield better performance?

2. The paper states that HuBERT achieves near-perfect recall but very low precision on CBR, suggesting it indiscriminately classifies most segments as commercials. Could the authors provide a more in-depth analysis of why this occurs? Is it due to the model's training objective, the nature of its learned representations, or a mismatch between its pre-training task and the fingerprinting task?

3. The segmentation algorithm uses a z-score threshold θ. The results in Table 7 show that smaller θ values (leading to shorter segments) yield better performance. Can the authors elaborate on the trade-offs of using very short segments?

---

> ### Author Response · Authors · 2025-11-17
> **Initial Response (1 of 2)**
>
> We appreciate your positive feedback!
> All your comments have been addressed, and the updated content is in the uploaded PDF.
> We are happy to answer any further questions.
>
> ---
>
> ### **Comment 1: discuss whether VLAFP's inference time is a bottleneck to practical use**
> > The paper notes that VLAFP has a longer inference time due to the overhead of handling variable-length segments. This could be a practical limitation for real-time applications.
>
> Thanks for raising this point.
> The slightly slower inference time of VLAFP **does not** pose a practical bottleneck.
> In Commercial-Broadcast Retrieval, commercials are fingerprinted offline, so no time constraints exist.
> During online broadcast monitoring, segmentation and fingerprinting run in parallel:
> - Segmentation determines when to form new segments from incoming audio frames.
> - Fingerprinting converts each segment into a fingerprint.
> With a minimum segment length of $T_{\min}=0.5$ s $= 500$ ms, segmentation inherently reuqires longer time than the fingerprinting process, since fingerprinting only takes 80 ms, ensuring that the fingerprinting process is always ready for the next segment without creating data congestion.
>
> While search efficiency could be further improved (e.g., through algorithmic optimization or additional computational resources), this is beyond the scope of our focus.
> **We have incorporated this explanation into the paper.**
>
> ---
>
> ### **Comment 2: discuss trade-off between accuracy and efficiency**
> > The trade-off between accuracy and efficiency is not deeply analyzed.
>
> We appreciate the suggestion and **have added a more detailed discussion to the paper**:
>
> - We compare with the AMG method, which achieves the lowest model size and training time, but exhibits significantly lower performance ($12\%-60\%$ worse across metrics).
> - We clarify that VLAFP achieves higher accuracy at the cost of slightly lower inference efficiency, primarily due to data loading overheads.
> - We discuss how the improved accuracy of VLAFP is supported by its fingerprint storage requirements.
>
> We hope this more detailed analysis provides a clearer picture of the trade-offs, and we would be happy to provide further insights if desired.
>
> ---
>
> ### **Comment 3: report performance on real-world broadcasts**
> > The audio augmentations (time-stretching, background noise, impulse response) are synthetically applied. While standard in the field, real-world broadcast distortions may be more complex and heterogeneous, and the robustness of VLAFP in truly uncontrolled environments remains to be validated.
>
> Thank you.
> As suggested, we have conducted new experiments on a real-world dataset named _Broadcast Audio Fingerprinting_.
> Our experiments use 135 reference (commercial) audio clips, along with TV-recorded broadcasts that contain these reference audios, which exhibit real-world distortions.
>
> |            | wav2vec2 | HuBERT | AST   | AMG   | NAFP  | **VLAFP (Ours)** |
> | ---------- | -------- | ------ | ----- | ----- | ----- | ---------------- |
> | F1 (%)     | 20.62    | 20.73  | 24.14 | 23.10 | 32.52 | **34.19**        |
>
> VLAFP still achieves the best performance which further validates the effectiveness and superiority of VLAFP under real-world distortions.
> **We have added the result table and more experimental details to the paper.**
>
> ---

---

> ### Author Response · Authors · 2025-11-17
> **Initial Response (2 of 2)**
>
> ### **Comment 4: describe baseline details**
> > Could the authors clarify the exact versions and whether they were fine-tuned for the fingerprinting task?
>
> Thank you for pointing this out.
> To ensure a fair comparison without cherry-picking, we used the model versions as provided on the Hugging Face model hub homepages:
> - wav2vec 2.0: _facebook/wav2vec2-base-960h_
> - HuBERT: _hubert-large-ls960-ft_
> - AST: _MIT/ast-finetuned-audioset-10-10-0.4593_
>
> We primarily follow the usage instructions of these baseline methods by using their implementations directly. However, we found the reviewer's comments insightful and are currently conducting additional experiments.
> **We have added these details to the paper for clarity.**
>
> ---
>
> ### **Comment 5: report performance on more baselines**
> > Would more recent self-supervised learning (SSL) models—such as WavLM or Whisper—yield better performance?
>
> As suggested, we have conducted additional experiments on a real-world dataset named _Broadcast Audio Fingerprinting_ and added WavLM and Whisper in the experiments.
> The best F1 achieved by each method is reported below.
> - WavLM: _microsoft/wavlm-base_
> - Whisper: _openai/whisper-base_
>
> |            | wav2vec2 | HuBERT | AST   | **WavLM** | **Whisper** | AMG   | NAFP  | **VLAFP (Ours)** |
> | ---------- | -------- | ------ | ----- | --------- | ----------- | ----- | ----- | ---------------- |
> | **F1 (%)** | 20.62    | 20.73  | 24.14 | 21.19     | 20.76       | 23.10 | 32.52 | **34.19**        |
>
>
> VLAFP still achieves the best performance.
> WavLM and Whisper behave similarly to other general audio representation baselines.
> **We have added the result table and more experimental details to the paper.**
>
> ---
>
> ### **Comment 6: explain the bad performance of baseline HuBERT**
> > The paper states that HuBERT achieves near-perfect recall but very low precision on CBR, suggesting it indiscriminately classifies most segments as commercials. Could the authors provide a more in-depth analysis of why this occurs? Is it due to the model's training objective, the nature of its learned representations, or a mismatch between its pre-training task and the fingerprinting task?
>
> We appreciate the insightful question.
> Our observations are as follows:
> - Different training objectives: HuBERT uses masked prediction leveraging contextual information without distortions, while our VLAFP is trained to minimize the distance between an audio segment and its distortions.
> - Different embedding characteristics: HuBERT’s high-dimensional embeddings may retain more high-level semantic information, while VLAFP produces compact, fingerprinting-specific representations.
> - Different temporal granularity: HuBERT may average out temporal details within its context windows, while VLAFP’s frame-to-segment design preserves temporal information crucial for fingerprinting.
>
> **We have incorporated this in-depth analysis in the paper.**
>
> ---
>
> ### **Comment 7: discuss trade-off of using very short segments**
> > The segmentation algorithm uses a z-score threshold θ. The results in Table 7 show that smaller θ values (leading to shorter segments) yield better performance. Can the authors elaborate on the trade-offs of using very short segments?
>
> Using a smaller threshold $\theta$ produces many more _shorter segments_.
> The advantages include:
> - More segments for training, meaning more training data points.
> - Finer-grained segment information, as the frame-to-segment stage preserves more frame-level details.
> - Improved performance, as observed in our study.
>
> The trade-offs are longer runtime and increased storage requirements. For example, the storage needed more than doubles when $\theta$ decreases from 2 to 0.
> **We have added this discussion to the paper**.

---

> > ### Comment · Area_Chair_grNQ · 2025-11-25
> >
> > Dear Reviewer Y1ti,
> >
> > The authors have responded to your reviews. Please review and provide your feedback and responses.
> >
> > Best,
> >
> > Your AC

---

> ### Comment · Reviewer_Y1ti · 2025-11-25
>
> Thank you for addressing my concerns, particularly the addition of WavLM and Whisper baselines. This significantly strengthens the empirical validation of VLAFP’s effectiveness. The new results demonstrate that VLAFP consistently outperforms even recent SSL models (WavLM: 21.19% F1, Whisper: 20.76% F1 vs. VLAFP’s 34.19% on FMA), reinforcing its superiority in handling variable-length semantics. Given this substantial improvement in experimental rigor and the robustness of the comparison, I am revising my overall rating from 6 to 8.

---

> > ### Author Response · Authors · 2025-11-25
> > **Initial Follow-up Response to Reviewer Y1ti**
> >
> > Thank you very much for your thoughtful response and for the score increase!
> > We sincerely appreciate your suggestion on comparing with WavLM and Whisper, as it further validates the strengths of our proposed VLAFP.

---

### Official Review · Reviewer_vupA · 2025-10-31

**Soundness:** 2
**Presentation:** 3
**Contribution:** 2
**Rating:** 2
**Confidence:** 4

**Summary:**

The paper extends the notion of fixed-length segment audio fingerprinting to **variable-length** segmentation. An audio recording is divided into variable-length segments determined by measures such as spectral entropy, and each segment is fed into a transformer-based encoder to obtain an embedding (fingerprint).
Training is performed by contrastive learning.
Experiments have been performed over speech, music, and acoustic events datasets.

**Strengths:**

### Originality
- The paper provides a clean extension of neural audio fingerprinting by replacing the convolutional encoder with a transformer encoder and by proposing a segment-level embedding refinement strategy.
- The model is trained using a standard contrastive loss as used in the baseline Neural Audio FingerPrinting (NAFP) [Chen et al.].
- The idea of segmenting database recordings into variable-length segments based on spectral entropy is conceptually interesting and practically motivated.
- While simple, the formulation is a logical step forward for audio fingerprinting systems.

### Clarity
- The presentation is clear and easy to follow.
- The description of the variable-length segmentation procedure and cross-attention refinement is well articulated.

### Significance
- The model introduces a new approach to neural audio fingerprinting and offers a straightforward way to handle variable-length segments.

**Weaknesses:**

### Conceptual
- The paper is primarily empirical, with limited theoretical analysis.
- The empirical analysis could be enhanced by including other tasks that rely on audio representation learning. Currently, the scope is narrowly focused on audio retrieval; broader implications for general-purpose audio representation learning are unexplored.
- The method restricts segment lengths to a predefined range \([T_{\min}, T_{\max}]\), raising a question about how queries shorter than \(T_{\min}\) could be handled. Additionally, events shorter than $T_{\min}$ may be merged with neighboring frames, potentially reducing fingerprint specificity.

### Experiments
- Another important metric to compare is storage efficiency (how much storage space the fingerprints require).
- The proposed method has a high hit rate, but is slower to search (Table 4). Could this be a bottleneck for practical use? I have a question here: What is the necessity of loading and locating audio when we are searching in the fingerprint database (Sec. 5.3)?
- Fig. 5: How is the hit rate close to $100\%$ with $\theta=\infty$? I think it will lead to segments of fixed length without any overlap, and hence, any relative time delay between the query segment start time and the target segment start time will result in the embeddings not matching.

### Presentation
- What is spectral entropy? Variable-length audio segmentation is an essential part of the contribution, but has been discussed very superficially in the paper.
- Fig. 6: In the comparison across different segmentation methods, it appears that the waveform method is better than spectral segmentation, although marginally, for many query durations. It would be beneficial if the paper also compared these methods in other ways, such as the nature of the segments obtained, to justify the choice of one method.
- At some places, the notations are written in a very non-standard way. E.g., in line 124, $A=A[1,2,...,N]$.
- Minor typos. E.g., line 023: advertisers has.


### Overall
The contribution looks less on the ML side and more suitable for an audio-focused venue such as INTERSPEECH, ISMIR, ICASSP, or IEEE TASLP.

**Questions:**

- Please address the questions mentioned in the Weaknesses section.
- Could you provide additional experiments demonstrating the model’s potential as a general-purpose acoustic representation learner (e.g., on non-retrieval audio benchmarks)?
- How does the method behave for queries shorter than the minimum segment length \(T_{\min}\)? Are they padded, merged, or discarded?

---

> ### Author Response · Authors · 2025-11-17
> **Initial Response (1 of 3)**
>
> Thank you for the thorough comments!
> We have addressed all your concerns below, and the revised manuscript with newly added content is included in the updated PDF.
> We welcome any further feedback or suggestions you may have!
>
> ---
>
> ### **Comment 1: empirical vs. theoretical**
> > The paper is primarily empirical, with limited theoretical analysis.
>
> Our work is motivated by real-world applications such as audio replay detection for commercial attribution and copyright protection. So the purpose of this paper is to present a new solution (VLAFP) to address practical challenges and deliver a comprehensive evaluation of VLAFP in realistic scenarios. Although the paper is mainly focused on empirical experiments, we think the new idea of variable length fingerprinting can also be a theoretical contribution.
>
> ---
>
> ### **Comment 2: report results on other tasks**
> > The empirical analysis could be enhanced by including other tasks that rely on audio representation learning. Could you provide additional experiments demonstrating the model’s potential as a general-purpose acoustic representation learner (e.g., on non-retrieval audio benchmarks)?
>
> Thank you for this insightful suggestion.
> Indeed, additional experiments on non-retrieval tasks would further demonstrate the model’s generalizability. To challenge VLAFP, we applied it to a very different task: keyword spotting, which requires more semantic similarity than the acoustic similarity that VLAFP is trained for.
> - The keyword spotting task using the _SpeechCommand v1_ dataset (containing 100k+ audios), adopting the _SUPERB benchmark_ setup.
> VLAFP achieves 40.00\% accuracy, while the audio representation models like wav2vec2-base and HuBERT-large reported over 90\% accuracy (96.23\% for wav2vec2-base and 95.29\% for HuBERT).
> - Due to time constraints, we only conducted zero-shot evaluation on the keyword spotting task. Since our training loss mainly enforces acoutsitc similarity rather than semantic similarity, the lower performance is justifiable.
> - We recognize the promising potential of extending VLAFP for broader audio tasks through classification-based objectives or fine-tuning strategies. Therefore, we are conducting experiments using task-specific losses and applying the same variable length method, and will follow this direction more in the future research.
>
> ---
>
> ### **Comment 3: describe how VLAFP handles shorter-than-500ms query segments**
> > The method restricts segment lengths to a predefined range [$T_{\min}$, $T_{\max}$], raising a question about how queries shorter than $T_{\min}$ could be handled. Additionally, events shorter than may be merged with neighboring frames, potentially reducing fingerprint specificity. How does the method behave for queries shorter than the minimum segment length $T_{\max}$? Are they padded, merged, or discarded?
>
> For further clarification, we have added more details and a concise pseudocode to describe the Variable-Length Segmentation algorithm.
> In general, $T_{\min}$ is an initialized frame number to form a segment. So for a very short event, it will be combined with other frames to form a short segment (for example, of length $T_{\min}$).
> Since $T_{\min}$ is a small value, it won't significantly reduce fingerprint specificity.
> Based on our statistics, these short segments account for less than $1.7\%$ of the total audio segments and do not affect the detection of significant presence of a commercial or target.
> **We have added the clarification to the paper.**

---

> ### Author Response · Authors · 2025-11-17
> **Initial Response (2 of 3)**
>
> ### **Comment 4: report storage efficiency**
> > How much storage space the fingerprints require?
>
>  As suggested, we have compared the required storage of raw fingerprints across methods for both the Commercial-Broadcast Retrieval (CBR) and Dummy-Target Retrieval (DTR) tasks on the FMA dataset:
> | Method               | dim \(d\) | CBR Commercial #seg | CBR Commercial size | CBR Broadcast #seg | CBR Broadcast size | DTR Dummy #seg | DTR Dummy size | DTR Target #seg | DTR Target size |
> |----------------------|-----------|-------------------|-------------------|------------------|------------------|----------------|----------------|----------------|----------------|
> | wav2vec2             | 1568      | 15k | 87M | 289k             | 1.7GB            | 581k           | 3.6GB           | 30k            | 177MB           |
> | HuBERT               | 1568      | 15k | 87M | 289k             | 1.7GB            | 581k           | 3.6GB           | 30k            | 177MB           |
> | AST                  | 527       | 15k | 29M | 289k             | 581MB            | 581k           | 1.3GB           | 30k            | 60MB            |
> | AMG                  | 128       | 15k | **7M** | 289k             | **141MB**        | 581k           | **299MB**       | 30k            | **15MB**        |
> | NAFP                 | 128       | 15k | **7M** | 289k             | **141MB**        | 581k           | **299MB**       | 30k            | **15MB**        |
> | **VLAFP (Ours)**     | 256       | **12k** | *12M*             | **220k**         | *214MB*          | 581k           | *597MB*         | 30k            | *29MB*          |
>
> - As expected, storage grows linearly with both the fingerprint dimension and the number of segments.
> For the CBR task, our spectral-entropy-based segmentation produces $20\%-24\%$ fewer segments, improving storage efficiency. For the DTR task, all methods use fixed-length 1-second segmentation, so the number of segments is the same (581k for Dummy and 30k for Target); in this case, storage is proportional to the fingerprint dimension.
> - We chose $d=256$ for our VLAFP because our hyperparameter study shows it achieves better performance.
> Nevertheless, storage efficiency can be easily improved by using a smaller $d$ (e.g., $d=128$) if desired.
>
> **We have added the storage table to the paper.**
>
> ---
>
> ### **Comment 5: discuss whether VLAFP's inference time is a bottleneck to practical use**
> > The proposed method has a high hit rate, but is slower to search (Table 4). Could this be a bottleneck for practical use?
>
> The slightly slower inference time of VLAFP **does not** pose a practical bottleneck.
> In Commercial-Broadcast Retrieval, commercials are fingerprinted offline, so no time constraints exist.
> During online broadcast monitoring, segmentation and fingerprinting run in parallel:
> - Segmentation determines when to form new segments from incoming audio frames.
> - Fingerprinting converts each segment into a fingerprint.
> With a minimum segment length of $T_{\min}=0.5$ s $= 500$ ms, segmentation inherently reuqires longer time than the fingerprinting process, since fingerprinting only takes 80 ms, ensuring that the fingerprinting process is always ready for the next segment without creating data congestion.
>
> While search efficiency could be further improved (e.g., through algorithmic optimization or additional computational resources), this is beyond the scope of our focus.
> **We have incorporated this explanation into the paper.**
>
> ---
>
> ### **Comment 6: explain the need of audio locating**
> > I have a question here: What is the necessity of loading and locating audio when we are searching in the fingerprint database (Sec. 5.3)?
>
> Thank you for your question—this pertains to an implementation detail. We utilize _PyTorch's DataLoader_ to handle audio segments efficiently during both training and retrieval:
> - For fixed-length segmentation, all segments are 1 second long. Each row in the dataloader corresponds directly to a single audio segment, which makes loading straightforward.
> - For variable-length segmentation, the situation is more complex because segments vary in duration (from 0.5 to 5 seconds). To maintain efficiency, we configure each row in the dataloader to be large enough to store the longest possible segment. Additionally, we apply data packing, where multiple shorter segments (e.g., 0.5-second segments) can be packed into a single row. Masks are used to indicate the boundaries of each segment within the row.
>
> This packing strategy minimizes loading overhead at the cost of needing to locate segments within the loaded data using masks. Although it adds a slight overhead, the overall efficiency gains during batch loading and processing justify this approach. Moreover, this is a detailed implementation in our experiments for simplicty, other methods/tools/optimizations can be applied to greatly improve the system efficiency in the deployment. **We’ve added clarification on this in the paper.**

---

> ### Author Response · Authors · 2025-11-17
> **Initial Response (3 of 3)**
>
> ### **Comment 7: explain the close-to-100% hit rate when $\theta=\infty$**
> > Fig. 5: How is the hit rate close to 100 with $\theta=\infty$? I think it will lead to segments of fixed length without any overlap, and hence, any relative time delay between the query segment start time and the target segment start time will result in the embeddings not matching.
>
>
> - First, we would like to clarify that the hit rate approaches 100% not because the threshold is infinite, but because the query length is long (increased to 10 seconds).
> - You are correct that with $\theta=\infty$, our VLAFP is trained using non-overlapping segments of length $T_{\max}=5$ seconds. However, note that Fig. 5 pertains to the Dummy-Target Retrieval task, where evaluation is based on one-second query segments.
> - More specifically, the near-100% hit rate shown corresponds to a query duration of 10 seconds. A 10-second query yields 19 overlapping one-second segments using a 0.5-second hop. Each of these segments is fingerprinted independently and contributes to retrieving the original audio. This means that even though VLAFP is trained on 5-second segments, it can still perform robustly on shorter, overlapping segments at test time.
> - It is also worth mentioning that for a given query length, an infinite threshold has never resulted in the best performance. Typically, the optimal threshold is around 1 across all query lengths.
> - Furthermore, since the method retrieves the most similar segment, it can tolerate some temporal misalignment as long as the correct segment ranks highest during retrieval.
>
> **We’ve added clarification on this behavior in the paper.**
>
> ---
>
> ### **Comment 8: describe the concept of _spectral entropy_**
> > What is spectral entropy?
>
> _Spectral entropy_ is an uncertainty measure in the distribution of spectral energy within an audio frame. To compute it, we apply a short-time Fourier transform to the audio signal to obtain its frequency energy distribution, normalize these energy values to form a probability distribution, and then compute the Shannon entropy of this distribution. Intuitively, a low spectral entropy indicates that the energy is concentrated in a narrow frequency range, as is typical of pure tones, whereas high spectral entropy reflects a more uniform spectral distribution, such as in white noise.
> **We have expanded the explanation of spectral entropy in the main text, and we now include a reference for its computation details.**
>
> ---
>
> ### **Comment 9: discuss the choice of segmentation method**
> > Fig. 6: In the comparison across different segmentation methods, it appears that the waveform method is better than spectral segmentation, although marginally, for many query durations. It would be beneficial if the paper also compared these methods in other ways, such as the nature of the segments obtained, to justify the choice of one method.
>
> Thank you for your thoughtful suggestion! *waveform* segmentation offers slightly better performance. However, it requires processing much more data points — for example, 8,000 values for just one second of audio at an 8kHz sampling rate. In contrast, our entropy-based approach reduces this to approximately 30 values. This corresponds to a 25x speedup. This efficiency also extends to the other entropy-based segmentation methods (PeLT and Silence).
> Given the strong performance and significantly efficient compute, we selected entropy-based segmentation as our primary method.
> **We have now clarified this rationale in the paper.**
>
> ---
>
> ### **Comment 10: notation and typos**
> > At some places, the notations are written in a very non-standard way. E.g., in line 124, $A=A[1,2,...,N]$; Minor typos. E.g., line 023: advertisers has.
>
> **We have replaced $A=A[1,2,...,N]$ with the more conventional $a=a[n]$ and corrected any found typo.**

---

> > ### Comment · Area_Chair_grNQ · 2025-11-25
> >
> > Dear Reviewer vupA,
> >
> > The authors have responded to your reviews. Please review and provide your feedback and responses.
> >
> > Best,
> >
> > Your AC

---

### Author Response · Authors · 2025-11-17

Dear Reviewers,

We sincerely appreciate the time, effort, and thoughtful feedback you have provided. We are delighted that our work has sparked interest and generated constructive questions. Especially questions on further exploration aligns perfectly with our goal: to introduce Variable-Length Audio FingerPrinting (VLAFP), a novel approach that not only addresses practical challenges in identifying audio replays, but also pioneers variable-length handling with potential applications in broader audio tasks such as keyword spotting and speaker identification.

We are grateful for your recognition of the strengths of our VLAFP, including its strong originality (Reviewers **vupA, Y1ti, 483Z**), clarity of presentation (**vupA, Y1ti, 483Z**), validated effectiveness (**Y1ti, 483Z**), and high significance (**vupA, Y1ti, 483Z**).

We have carefully considered all concerns raised and have addressed each point in our individual responses. In particular, we have revised the paper to clarify and strengthen the description of our method, provide additional experimental results, and better articulate the broader implications of our approach. All changes are clearly highlighted in the manuscript for your convenience: **blue** text indicates newly added content, and **red** text indicates modifications to existing text.

We would also like to reiterate the key contributions and impact of our work:
- To the best of our knowledge, VLAFP is the first deep learning-based variable-length audio fingerprinting method, paired with a variable-length segmentation approach, designed to overcome limitations of fixed-length fingerprinting.
- Extensive experiments demonstrate that VLAFP consistently outperforms existing methods on live audio identification and offline audio retrieval across three real-world datasets.
- Our approach opens multiple avenues for future research, including exploration of segmentation strategies, data augmentation techniques, and self-supervised loss functions, further highlighting the method’s broader significance.

We hope that the clarifications, additional experiments, and detailed responses provided demonstrate the rigor, novelty, and potential impact of our work. We remain happy to address any further questions or feedback you may have.

Thank you once again for your time, thoughtful comments, and consideration.

---

### Comment · Area_Chair_grNQ · 2025-11-22
**Official Comment by AC**

Dear Authors and Reviewers,

I would like to thank the authors for providing detailed rebuttal messages on time.

To reviewers: I would like to encourage you to carefully read all other reviews and the author responses and engage in an open exchange with the authors. Please post your first response as soon as possible within the discussion time window. Ideally, all reviewers will respond to the authors, so that the authors know their rebuttal has been read.

Best regards,
AC

---

> ### Author Response · Authors · 2025-11-22
>
> Dear AC,
>
> Thank you, we appreciate your following up on our responses!
>
> Best Regards,
>
> ---
>
> Dear Reviewers,
>
> We have tried our best to thoroughly address your initial comments. We would love to hear whether you have further concerns.
> Please let us know. Thanks again.
>
> Best Regards,

---

### Comment · Area_Chair_grNQ · 2025-11-27

Dear Reviewers,

Thank you for your valuable reviews. With the Reviewer-Author Discussions deadline approaching, please take a moment to read the authors' rebuttal and the other reviewers' feedback, and participate in the discussions and respond to the authors. Finally, be sure to complete the "Final Justification" text box and update your "Rating" as needed. Your contribution is greatly appreciated. I will flag irresponsible (final) reviews and/or any reviewers not participating in discussions.

Reviewers are expected to stay engaged in discussions, initiate them and respond to authors’ rebuttal, ask questions and listen to answers to help clarify remaining issues.

It is not OK to stay quiet.

It is not OK to leave discussions till the last moment.

If authors have resolved your (rebuttal) questions, do tell them so.

If authors have not resolved your (rebuttal) questions, do tell them so too.

Thanks.

AC

---

### Meta-Review · Area_Chair_Xjub · 2026-01-06

**Summary:**

The main concerns are around limited novelty, lack of comparisons against strong SSL baselines, technical clarity. The authors provided a comprehensive rebuttal with new experiments, new datasets, stronger baselines. However, the novelty seems limited, as suggested by vupA, maybe a speech focused venue is more suitable.

**Reviewer Concerns:**

The following concerns are properly addressed:

vupA:
* requested on general purpose representation learning tasks: the authors added key-word spotting tasks
* efficiency: the author added new results
* under-explained concept, "spectral entropy" etc: the author added references for those
* edge case on min segment length: the author added more clarification to the paper.

Ylti:
* baseline not strong: authors added new baselines
* evaluation on real-world data: new results on Broadcast Audio Fingerprinting data were added
* bad performance on HuBERT: the authors argued due to different training objectives, embedding characteristics and temporal granularity

3yp7:
* math notation errors: fixed in the revision
* challenged the claim "loss of natural boundaries" as unsubstantiated: explained the misinterpretation

483Z:
* semantic segmentation: added comparisons
* formatting: adjusted

The following concerns are not fully addressed:

vupA:
* too empirically focused: the rebuttal is not convincing enough

3yp7:
* limited novelty

483Z:
* segmentation threshold is tuned data dependent: the authors acknowledged and will explore in future work

**Reviewer Scores:**

vupA: maintained 2 during discussion, no change
Ylti: initial 6, updated to 8, no further change
3yp7: 2, no further change
483Z: 6, no further change

---

### Decision · Program_Chairs · 2026-01-26

Reject